# Magnetoencephalography recordings reveal the neural mechanisms of auditory contributions to improved visual detection

Alexis Pérez-Bellido [1,2✉], Eelke Spaak[1] & Floris P. de Lange [1]

Sounds enhance the detection of visual stimuli while concurrently biasing an observer's decisions. To investigate the neural mechanisms that underlie such multisensory interactions, we decoded time-resolved Signal Detection Theory sensitivity and criterion parameters from magneto-encephalographic recordings of participants that performed a visual detection task. We found that sounds improved visual detection sensitivity by enhancing the accumulation and maintenance of perceptual evidence over time. Meanwhile, criterion decoding analyses revealed that sounds induced brain activity patterns that resembled the patterns evoked by an actual visual stimulus. These two complementary mechanisms of audiovisual interplay differed in terms of their automaticity: Whereas the sound-induced enhancement in visual sensitivity depended on participants being actively engaged in a detection task, we found that sounds activated the visual cortex irrespective of task demands, potentially inducing visual illusory percepts. These results challenge the classical assumption that sound-induced increases in false alarms exclusively correspond to decision-level biases.

[1] Donders Institute, Radboud University, Nijmegen, The Netherlands. [2] Department of Cognition, Development and Educational Psychology, University of Barcelona, Barcelona, Spain. ✉email: alexisperezbellido@gmail.com

Although humans largely rely on vision in order to monitor the environment[1,2], our brain has learnt that sensory signals from external events are often correlated across sensory modalities and should be optimally integrated[3,4]. In the last years multiple studies have revealed that regions classically considered as visual areas are tuned to sensory information conveyed by both visual and auditory signals[5–8]. These results suggest that the processing of visual information might be modulated by auditory inputs already at the earliest perceptual stages[9]. In visual detection, previous behavioral research has shown that synchronizing a sound with a visual stimulus improves the detection threshold of the latter. This phenomenon, termed as the sound-induced visual enhancement, has been described using the Signal Detection Theory framework (SDT)[10] in multiple psychophysical studies[11–13]. While task-irrelevant sounds improve participants sensitivity (d') in discriminating visual targets from noise, they also increase the proportion of reported false alarms, leading to a reduction in the criterion parameter (c)[11,13–16]. However, although the behavioral consequences of sounds on visual detection have been well characterized psychophysically, we still lack a good description of which specific neural mechanisms underlie such crossmodal decision-making detection biases.

Most previous neuroimaging research has focused on understanding how sounds lead to visual enhancements in detection. Nowadays, it is widely accepted that sounds modulate the processing of visual signals at early perceptual stages through sensory-level interaction of audiovisual inputs[8,17–20]. Yet, more recent studies tackling similar questions from a perceptual decision-making perspective[21,22] have characterized the effect of sounds on visual discrimination as a late enhancement in the transformation of sensory input into accumulated decisional evidence. Therefore, how specifically and to which extent sounds enhance visual sensitivity at early (i.e., sensory encoding), late (i.e., decision formation) or both[17] perceptual stages remains under debate.

An equally relevant question that has received much less attention in perceptual neuroscience is how sounds induce changes in visual detection criterion. Although SDT criterion variations are typically interpreted as decision-level response biases (i.e., a stimulus-independent bias to report a given response), some perceptual illusions that affect perceptual accuracy often manifest in a shifted criterion parameter[23,24]. Thus, whether the sound-induced increase in false alarms in visual detection tasks exclusively represents a decisional-level bias or it might also reflect a perceptual-level bias is still unknown.

Finally, another disputed question taps into the automaticity of multisensory interplay[25–28]. While there is ample evidence that sounds can modulate neural activity in early visual areas automatically[6,7,29–31], several studies have shown that crossmodal interactions are weakened when attention is directed away from the relevant stimuli[32–36], or the sensory signals fall below the threshold of awareness[37,38]. Thus, whether and how sounds modulate the processing of synchronous but task-irrelevant visual information remains debated.

We addressed these questions in two different experiments: In the first experiment we tested a group of participants in a visual detection task while we concurrently registered their magnetoencephalographic (MEG) activity. Unlike previous studies in which the effect of sounds on visual processing was characterized using univariate contrasts[18,20,33,39,40], here we capitalized on multivariate analyses to decode time-resolved d' and c parameters from MEG activity patterns. By contrasting the decoders performance at different time points in the visual and audiovisual conditions, we characterized how sounds changed participants visual detection sensitivity and decision criterion over time.

Moreover, to test whether sounds simply modulate the neural representation of visual information at specific time points or they also affect its temporal persistence, we implemented temporal generalization (TG) analyses[41,42].

Because the goal of this study was to characterize how sounds modulate the neural processes underlying changes in visual sensitivity and decision criterion, we constrained our decoding analyses to four source-reconstructed brain regions of interest (ROIs) engaged in the broadcasting and maintenance of stimulus information during visual detection[43,44]. These were the early visual cortex, that is involved in the encoding of sensory information[45], the parietal cortex, that is related to evidence accumulation during perceptual decisions[46,47], the inferotemporal cortex, that has been associated to visual memory and target identification[48–50], and the dorsolateral prefrontal cortex (DLPFC), that is involved with short-memory, decision-making and awareness[51–54]. Finally, to explore if sounds modulate the rate of evidence accumulation during visual detection we supplemented our decoding analyses with a univariate sensor-level analysis of centroparietal activity[21,55].

In a second experiment we sought to understand whether sounds modulate the processing of visual stimuli automatically or through a top-down controlled mechanism. To do that, we tested a new group of participants with similar stimuli sequences as in the first experiment, but this time their attention was diverted away from the previously relevant visual gratings (Fig. 1c). Building on the same decoding methodology that we used for the first experiment, we investigated whether sounds changed the neural representations of unattended visual gratings across the same hierarchy of brain regions.

Foreshadowing our results below, we show that auditory input interacts with visual processing in two distinct ways: (1) Sounds enhance the detection of visual targets, primarily by improving the accumulation and maintenance of perceptual evidence across the visual hierarchy. This enhancement seems to be mediated by a top-down controlled mechanism, as it depends on the relevance of the audiovisual stimuli for the task; and (2) Sounds automatically trigger patterns of activity in the visual cortex highly similar to the ones evoked by a visual stimulus. This result suggests that the typically reported sound-induced behavioral increment in false alarms in visual detection tasks, instead of merely reflecting a decisional bias[11,13,16], it is potentially related to a bottom-up auditory-driven perceptual bias.

## Results

### Experiment 1: Testing the neural mechanisms that support the sound-induced visual enhancement.
Twenty-four participants underwent an MEG scan while they performed a visual detection task (Fig. 1c). In each trial, participants had to report the presence (S+) or absence (S−) of a briefly flashed vertical grating (33 ms). In half of the trials and orthogonal to the probability of appearance of the visual grating, an auditory stimulus was presented (the 'audiovisual' condition). Visual stimuli could be presented at the center (i.e., the fovea and parafovea) or the periphery (i.e., perifovea) of the visual field (Fig. 1c). This manipulation was motivated by previous neuroanatomical tracing studies in monkeys showing that more peripheral visual eccentricities receive denser projections from primary auditory cortex[56,57], and it allowed us to assess whether the mechanisms involved in audiovisual interaction depended on visual eccentricity[12,58].

### Experiment 1: Sounds improve visual detection sensitivity and reduce criterion.
Observer's performance was more accurate in the audiovisual compared to the visual conditions (P(Correct) = 0.73 vs. 0.69: $F_{1,23} = 22.5$, $P < 0.001$, $\eta = 0.09$; Fig. 2a). Next, we applied

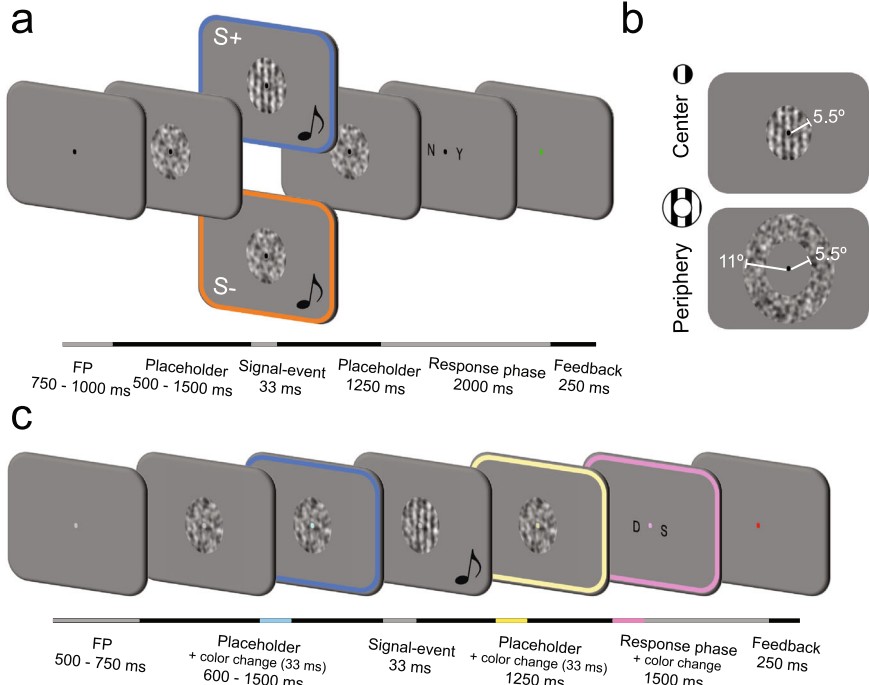

**Fig. 1 Experimental design description. a** Experiment 1 design: Trial sequence showing an audiovisual signal-present (S+) or signal-absent (S−) event at the central visual field. In the audiovisual trials (50% of the trials), a 1000 Hz pure tone was presented simultaneously (33 ms) with the [S+|S−] event (S + trials = 50% and their probability of appearance was decorrelated to the sound presentation). **b** Example of a central S+ and a peripheral S− stimulus. Information about their dimensions is overlaid in white. **c** Experiment 2 design: The trial sequence was similar to the one used in the first experiment with the difference that now participants had to ignore the visual grating and auditory events and report on fixation point (FP) color changes. For illustrative purposes, the color frames in (**a**, **c**) highlight those events that were task-relevant in each experiment.

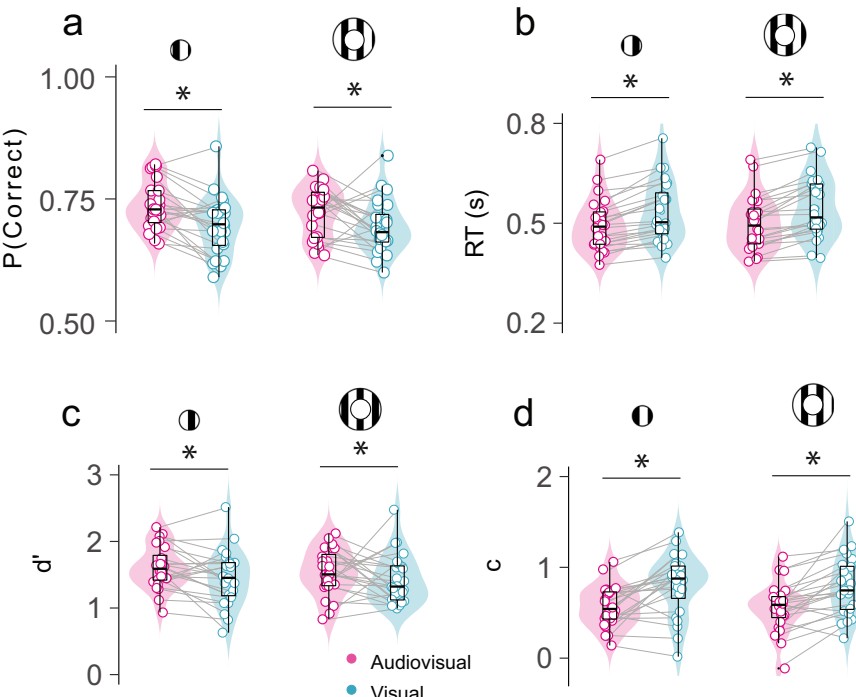

**Fig. 2 Sounds modulate participants visual detection performance.** Accuracy (**a**), reaction times (**b**), sensitivity (**c**) and criterion (**d**) are depicted for each modality (audiovisual/visual), visual eccentricity condition (center/periphery) and participant. Inside the violin plot, the horizontal black line reflects the median, the thick box indicates quartiles, and whiskers 2 times the interquartile range. Gray horizontal lines connect participants scores across conditions. Asterisks indicate significant differences between the visual and audiovisual conditions ($P < 0.05$).

SDT to disentangle the contribution of sounds to perceptual sensitivity and decision criterion. Participants' sensitivity ($d'$; see Methods) in detecting the visual target was higher in the audiovisual compared to the visual conditions ($d' = 1.57$ vs. $1.44$: $F_{1,23} = 5.67$, $P = 0.025$, $\eta = 0.032$; Fig. 2c). This sensitivity enhancement did not depend upon visual eccentricity (interaction between modality and eccentricity: $F_{1,23} = 0.16$, $P = 0.69$, $BF = 0.23$). Furthermore, the criterion (c) parameter was larger than 0 in both the visual and audiovisual conditions, manifesting an overall bias to classify the trials as signal-absent. However, as predicted, this bias was lower in the audiovisual compared to the visual conditions ($c = 0.56$ vs. $0.79$: $F_{1,23} = 23.97$, $P < 0.001$, $\eta = 0.014$; Fig. 2d), indicating that participants were more likely to report the signal presence in the audiovisual trials. Criterion did not differ between visual eccentricities ($F_{1,23} = 0.15$, $P = 0.7$, $BF = 0.23$) and the auditory-driven reduction in criterion did not depend on the visual eccentricity either ($F_{1,23} = 0.01$, $P = 0.9$, $BF = 0.21$). Finally, although participants were instructed to prioritize accuracy over speed, reaction times (RTs) were faster in the audiovisual than in the visual conditions (RTs = 496 vs. 540 ms: $F_{1,23} = 61.02$, $P < 0.001$, $\eta = 0.06$; Fig. 2c). Also for reaction times, the effect of modality did not depend upon visual eccentricity ($F_{1,23} = 0.24$, $P = 0.62$, $BF = 0.23$). Thus, our behavioral results are consistent with previous reports in showing that sounds enhance visual sensitivity, speed up visual detection and increase the proportion of reported false alarms (Supplementary Fig. 1). Given that we were not able to detect any significant effect of visual eccentricity on the sound-induced visual enhancement, in the following multivariate decoding analyses the center and periphery visual field conditions will be combined to train and test the classifier. Combining these two conditions will improve the sensitivity of the classifier (as the number of trials in each training fold will be doubled). As a sanity check, we performed a control analysis where we trained and tested the classifier with data from the center and periphery conditions separately and obtained qualitatively similar results in both visual fields (Supplementary Fig. 2).

**Experiment 1: Neurally decoded signal detection parameters correlate with behavior.** To describe how sounds modulate visual processing across the perceptual hierarchy we trained a linear discriminant classifier to distinguish signal-present (S+) from signal-absent (S−) trials on the basis of MEG signals measured in visual, parietal, inferotemporal and prefrontal ROIs (upper part, Fig. 3a). These ROIs were anatomically defined and encompassed multiple sensory and decision-related brain regions involved in visual detection[43,51]. Prior to testing the effect of sounds on visual detection, we explored how the neurally estimated $d'$ and c parameters evolved in time in the visual and audiovisual conditions together. We found that a classifier trained and tested from −0.2 to 1.5 s with respect to the stimulus onset was able to discriminate above chance level between S+ and S− trials from 180 ms to the end of the trial in the visual ROI ($P < 0.001$, $t_{max} = 14.3$, $t_{clust} = 4944$; black contours in Fig. 3c). A closer inspection of the $d'$ temporal generalization matrixes (TGMs) showed that signal decoding peaked around 500 ms in all the ROIs (Supplementary Fig. 3c and Fig. 3) and in consistence with previous research[43,59], the information encoded at early sensory stages (<250 ms) generalized less than at later decision-related stages. We repeated the same analysis on the neural decoder's criterion (c). We observed that criterion increased at 200 ms after stimulus onset and on both sides of the diagonal. The same pattern was replicated across ROIs (Supplementary Fig. 4). This result indicates that whereas a classifier trained to classify S+ and S− trials before 200 ms cannot be biased (given that there is not reliable information yet to be trained on), once the classifier

learns the difference between S+ and S− trials (after 200 ms until the end of the trial), it will classify more often the activity in those time points not containing decodable information as signal-absent, leading to larger criterion values.

Next, we tested whether the neurally derived SDT parameters encoded time specific information about the participant's latent perceptual decision-making states. We found that the neural decoder's $d'$ parameters were positively correlated with the observer's $d'$ along the TGM diagonal in all the ROIs (visual ROI: $P < 0.001$, $t_{max} = 22.1$, $t_{clust} = 4944$), peaking at 500 ms (see ROI averaged results in Fig. 3c), with subtle differences in the generalization spread between ROIs. Further correlation analyses revealed that the neural criterion parameters were positively correlated with the observer's criterion parameters in a first cluster spanning from 400 ms to 600 ms in visual and parietal ROIs ($P < 0.01$, $t_{max} = 5.6$, $t_{clust} = 183$; $P < 0.005$, $t_{max} = 5.4$, $t_{clust} = 88.5$), and in a second cluster spanning from 900 ms to 1100 ms in the visual, parietal and inferotemporal ROIs ($P < 0.01$, $t_{max} = 5.1$, $t_{clust} = 181$; $P < 0.05$, $t_{max} = 5.15$, $t_{clust} = 35.4$; $P < 0.001$, $t_{max} = 5.8$, $t_{clust} = 346$). These results demonstrate that the neural decoder's $d'$ and c parameters are informative about participants behavior and allow us to infer at which temporal latencies the brain performs critical sensitivity and bias computations.

**Experiment 1: Sounds enhance the maintenance of post-sensory visual information over time.** To understand how sounds neurally enhance visual detection we contrasted the audiovisual and visual $d'$ TGMs in each ROI (Fig. 4c). Four significant clusters conforming a continuous pattern ($P < 0.03$, $t_{max} = 5.75$, $t_{clust} = 92.5$) emerged in the inferotemporal ROI (Fig. 4b), meaning that the information decodable at 500 ms was better preserved until the end of the trial in the audiovisual condition. Interestingly, in the visual cortex ROI we found that decoders trained before and after the response time (1100 ms to the end of the trial) and tested in the audiovisual condition were more sensitive to information present at around 500 ms ($P < 0.02$, $t_{max} = 5.6$, $t_{clust} = 114.5$) than the same decoders tested in the visual condition. This result suggests that although visual target representations at 500 ms were steadily maintained in the visual and audiovisual conditions (Fig. 4a), the information was better preserved in the audiovisual trials, reaching the maximal decoding difference at the time in which participants were instructed to select their final response (1250 s). In the parietal cortex ROI we also observed a significant cluster ($P < 0.01$, $t_{max} = 4.9$, $t_{clust} = 31.48$) reflecting that sounds punctually boost signal decoding at around 500 ms in comparison to the visual trials. This effect was time specific as it did not generalize to earlier or later time points. We registered a late but brief (approximately from 1000 to 1100 ms) enhanced reinstatement of the information presented at around 500 ms in the dorsolateral prefrontal cortex ROI. This small cluster ($P < 0.02$, $t_{max} = 5$, $t_{clust} = 22.73$) preceded in time the cluster previously reported in the visual ROI (Supplementary Fig. 5).

To obtain a general description of the TG pattern, we looked at the $d'$ TGM averaged over ROIs. Consistently with the individual ROIs decoding analyses we found that the audiovisual enhancement of information conformed a continuous pattern (Fig. 4c). Decoders trained from 500 ms to the end of the trial could decode better the visual stimulus at 500 ms in the audiovisual compared to the visual condition. This vertical pattern was composed by two elongated clusters ($P < 0.001$, $t_{max} = 5$, $t_{clust} = 700.2$ and $P < 0.005$, $t_{max} = 4.8$, $t_{clust} = 245$) separated by a small gap spanning from approximately 750 to 950 ms where the sound-induced visual decoding enhancement was not significant.

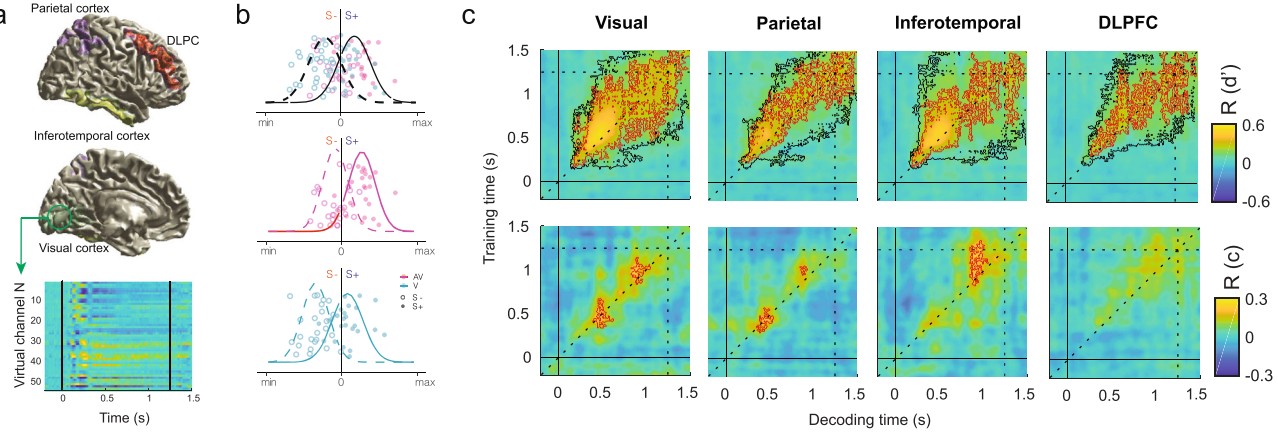

**Fig. 3 Decoding sensitivity and criterion parameters from neural activity patterns. a** Visual representation of the four anatomically defined regions of interest overlayed on a hemisphere surface model of the MNI template. Source-reconstructed activity extracted from the virtual channels that conform each ROI (e.g., visual cortex) is used to test and train a multivariate classifier (bottom part). **b** A multivariate pattern classifier was trained on visual and audiovisual (green and magenta simulated points respectively) conditions together to optimally discriminate between S+ and S− trials (empty and full circles respectively). The trained classifier was cross-validated on visual and audiovisual trials separately. The solid and dashed lines represent the probability density functions of the S+ and S− trials distributions based on the discriminant channel output. Specifically, trials with a discriminant channel output larger/smaller than 0 (vertical line) are classified as signal-present/absent respectively. We categorized S+ trials classified as signal-present as hits, and the S− trials classified as signal-present as false alarms. Note that although a classifier might be equally good in discriminating S+ from S− trials in all the conditions, there might be systematic biases to classify more often the trials in the audiovisual condition (middle panel) as signal-present than in the visual condition (bottom panel). **c** Correlation between behaviorally estimated and neurally decoded sensitivity (upper panels) and criterion (bottom panels) parameters. Red thin contours depict significant correlation clusters and black thin contours identify clusters in which the stimulus decoding sensitivity was larger than 0.

Surprisingly, we found that although multiple decoders trained at different time points could decode the signal presence at 500 ms, a decoder trained at 500 ms was not able to symmetrically enhance the decoding of the audiovisual trials within an analogous temporal window. In a complementary control analysis (Fig. 4c and Supplementary Fig. 6), we confirmed that by training and testing the classifier in the visual and audiovisual conditions separately, the sound-induced d′ enhancement pattern became symmetrical with respect to the diagonal ($P < 0.02$, $t_{max} = 4.9$, $t_{clust} = 296.5$). This result indicates that training the classifier in the visual and audiovisual conditions together might come with the cost of losing/averaging out the idiosyncratic features that characterize the visual and audiovisual neural activity patterns. Therefore, a classifier trained on data from both conditions together at the decoding peak (500 ms), but tested in the visual and audiovisual conditions separately might have reduced capacity to generalize the information and tease apart S+ and S− trials when the most discriminative information decays (i.e., before and after the decoding peak), and other modality-specific neural modulations add variability to the signal.

In summary, our d′ analyses show that sounds enhance the maintenance over time of late perceptual information (500 ms) until the response phase. This information maintenance is mediated through the functional interplay of the different ROIs.

**Experiment 1: Sounds enhance the gain of post-sensory evidence accumulation.** The d′ parameter decoding analyses showed that sounds enhanced the maintenance of late perceptual representations. This enhancement might be preceded by a sound-induced improvement in evidence accumulation[21]. This interpretation is supported by the sound-induced enhancement in information decoding in the parietal cortex (Fig. 4b). In contrast with previous multisensory research that characterized the activity in parietal regions as a signature of multisensory integration[60,61], in the light of our visual detection paradigm it might better reflect a change in the process of evidence

accumulation[55,62]. To test this hypothesis, we contrasted the unfolding of centroparietal event-related field (ERF) activity in the visual and the audiovisual conditions[55,62]. Thus, if sounds enhanced the accumulation of perceptual evidence, we expected to find a larger positive modulation of centroparietal activity in the audiovisual compared to the visual trials.

We first characterized the scalp topography of the (planar-combined) ERF activity evoked by the visual targets at the decoding peak (500 ms) in the visual and audiovisual conditions. Using cluster-based permutation (correcting for multiple comparisons across time) we found that whereas in the visual condition, the processing of the visual target induced a broadly spatially distributed activity pattern across the scalp topography (Fig. 5a), the processing of the visual target in the audiovisual condition was spatially constrained to visual and centroparietal sensors. To better understand how sounds modulated centroparietal activity, we limited our analyses to a subset of MEG sensors localized in centroparietal regions (see methods). This analysis showed that centroparietal activity unfolds very differently in the visual and audiovisual conditions: Whereas centroparietal activity increased slow but steadily in the visual condition ($P < 0.02$, $t_{max} = 5.01$, $t_{clust} = 11$), in the audiovisual condition the activity was initially supressed (at around 350 ms), and then it quickly ramped up ($P < 0.005$, $t_{max} = 3.9$, $t_{clust} = 39.2$, Fig. 5b). To quantify the differences in evidence accumulation rate between both conditions, we calculated the slope of the centroparietal activity modulation as a function of time using a 200 s ms sliding window (Fig. 5c). In consistence with the previous analysis, we observed that the activity gain from 250 to 350 ms was significantly reduced in the audiovisual compared to the visual trials ($P < 0.001$, $t_{max} = 4.49$, $t_{clust} = 91$), however, from 350 to 500 ms the activity gain was sharply reversed experiencing a strong increment compared to the visual condition ($P < 0.05$, $t_{min} = -3.29$, $t_{clust} = -43$). These results demonstrate that evidence accumulation in the visual condition unfolds more gradually than in the audiovisual condition. However, the integration of evidence in the audiovisual conditions evolves

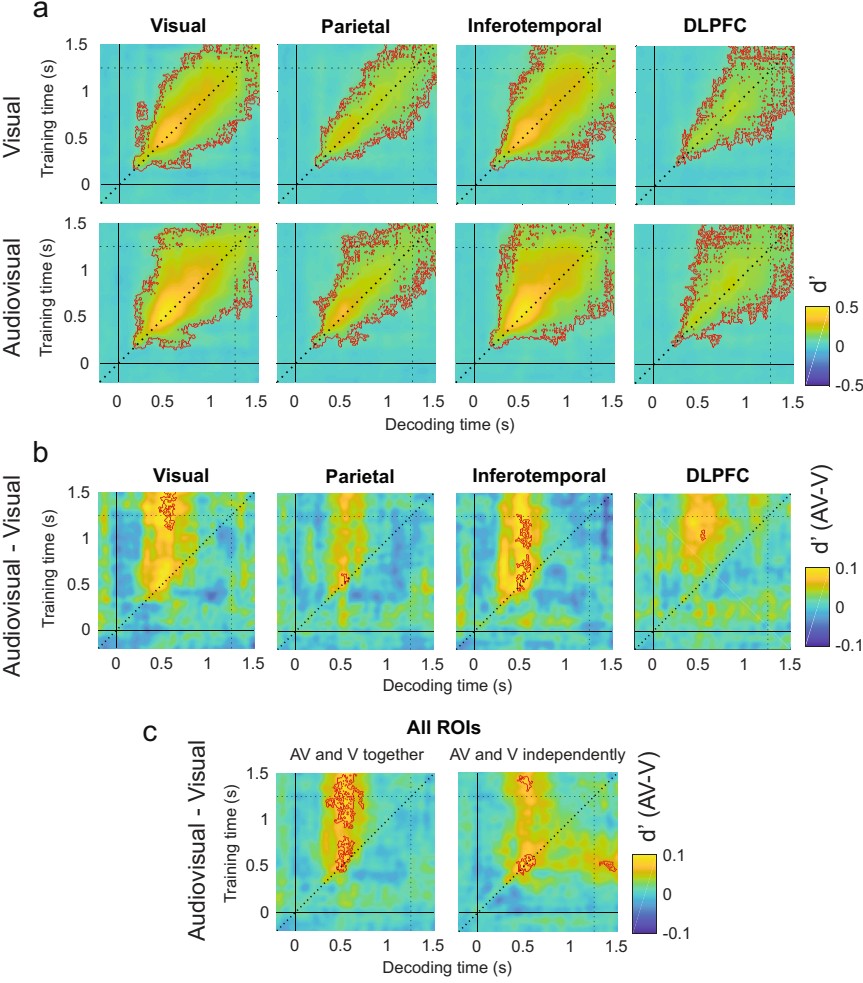

**Fig. 4 Visual sensitivity decoded from visual and audiovisual stimuli across ROIs.** Temporal generalization matrixes depict (**a**) the neural decoder's sensitivity parameter (d′) in the visual (upper TGMs) and audiovisual (bottom TGMs) conditions and their difference (**b**) in each ROI. **c** Represents the differential decoded sensitivity averaged over ROIs with classifiers trained in the visual and audiovisual conditions together (left panel) or separately (right panel). Independently trained classifiers replicate similar results as when the classifiers were trained in both conditions together and recovers the expected decoding symmetry with respect to the diagonal. Red contours delimit significant clusters. Vertical and horizontal dotted lines represent the response phase onset.

later but much faster. It is possible that sounds helped the participants to condense their processing resources on those temporal latencies in which the target information is more readily available, boosting the efficiency of the evidence accumulation process and subsequently leading to a better encoding and maintenance of decision information. This hypothesis is supported by the scalp topography in the audiovisual condition that in contrast with the visual condition, revealed a more efficient recruitment of those brain regions specialized in perceptual decision-making (i.e., the visual and parietal cortices).

**Experiment 1: Neurally decoded criterion is lower in the audiovisual trials.** Consistent with participants behavioral performance, we found that the neural decoder's criterion was reduced in the audiovisual compared to the visual condition in all the ROIs (Fig. 6a). That is, a decoder trained on visual and audiovisual trials together systematically classified more often the audiovisual trials as S+ trials than the visual ones (Supplementary Fig. 7). This pattern was predominant along the TGM diagonals with some qualitative differences between ROIs (Fig. 6b). In the visual and inferotemporal cortex ROIs we observed significantly

negative clusters ($P < 0.005$, $t_{max} = 5.6$, $t_{clust} = 114$; $P < 0.02$, $t_{max} = 5.6$, $t_{clust} = 52$), indicating that multiple decoders trained from 500 to 1100 ms were biased to classify audiovisual trials patterns occurring at 400 ms as S+ trials. In the parietal and dorsolateral prefrontal ROIs, a similar but less generalized pattern emerged along the diagonal ($P < 0.01$, $t_{max} = 5.7$, $t_{clust} = 54$; $P < 0.02$, $t_{max} = 5.06$, $t_{clust} = 6.8$). In addition, we observed a second significant cluster that was specific to the parietal and dorsolateral ROIs. In this second cluster, the direction of the bias reversed. That is, a decoder trained around 400 ms was biased to classify more often a trial as S+ in the visual than in the audiovisual trials (from 600 to 1100 ms in the parietal ROI; $P < 0.01$, $t_{max} = 4.9$, $t_{clust} = 31.4$, and from 1000 to 1100 ms in the dorsolateral ROI; $P < 0.02$, $t_{max} = 5.06$, $t_{clust} = 22.7$). The latency of this positive bias may indicate that whereas in the S− audiovisual trials the participants believed more consistently to have seen the target immediately after the sound presentation, in the S- visual trials, due to the absence of an auditory cue signaling the most likely latency of the visual-stimulus onset, the belief of seeing the target can potentially take place at any other time point.

In summary, criterion analyses demonstrate that sounds evoke neural activations that are more often misclassified as S+ trials

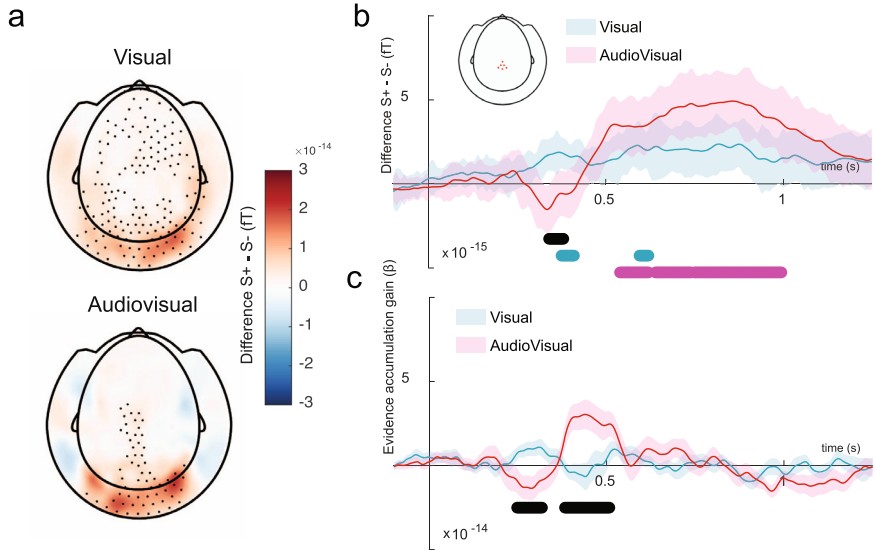

**Fig. 5 Sounds boost the gain of post-sensory evidence accumulation.** a Group-level difference difference between the sensor-level ERF activity in the signal-present and signal-absent conditions measured at (**a**) whole-brain scalp topography, and (**b**) centroparietal sensors (red points in the topography inset). **c** Evidence accumulation gain, measured as the slope of the differential ERF activity in B, as a function of time using a 200 ms sliding window. The black points in a depict those MEG sensors with significantly different activity in the S+ and S− conditions. Line contours depict standard error. Magenta and green rectangles compress temporal clusters in which there are significant differences between the S+ and S− conditions. Black rectangles depict temporal clusters in which there are significant differences between the audiovisual and visual timeseries. Group-level statistics were obtained using cluster-based permutation tests.

than in the visual conditions. As expected, in a control analysis we showed that this sound-induced criterion bias would be inappreciable if we would have trained and tested the decoders in the visual and audiovisual conditions independently (Fig. 6c).

**Experiment 2: Assessing the automaticity of audiovisual interaction.** In our second experiment we tested if the mechanism by which sounds enhanced the maintenance of visual information is automatic or requires participants to be actively engaged in a visual detection task. For instance, in line with the previous analyses on evidence accumulation (Fig. 5), it is possible that subjects used the auditory stimulus as a cue to optimally orient their attention in time, prioritizing the encoding and maintenance of the most informative visual samples in short-term memory[13,63,64]. Such a mechanism cannot be considered bottom-up as it requires an endogenous control of attention. Relatedly, we also investigated whether the reduced criterion in the audiovisual condition reflects a decisional or a perceptual bias.

To address these questions, a new group of participants ($n = 24$) was presented with similar stimuli sequences as in our first experiment. However, participants now had to attend to and memorize a sequence of three fixation point color changes (Fig. 1c), rendering the previously relevant vertical gratings and auditory stimuli task irrelevant. The unattended gratings were presented at only one visual eccentricity (central) and with two different levels of contrast (i.e., contrast was set at threshold level in the low contrast S+ condition and at two-times the threshold level in the high contrast S++ condition). Adding this new high contrast condition allowed us to test whether perceptually salient but ignored S++ stimuli interact with sounds, and ensured that unattended visual stimuli evoked sufficient signal to be decoded from neural activity patterns.

We hypothesized that (1) if the enhanced maintenance of neurally decoded visual information described in our first experiment is supported by automatic bottom-up multisensory interaction, we should replicate the same result in this second experiment despite participants ignore the audiovisual stimuli.

Following up on the same logic, (2) if the reduction in the decoder's criterion following the sound presentation that we described in the first experiment corresponds to a decision-level bias, we should not replicate here the same result, as participants do not make decisions about the visual gratings. Instead, finding again a more biased decoder in the audiovisual condition would imply that sounds automatically activate the brain, evoking activity patterns that are misclassified by the decoder as a visual stimulus.

**Experiment 2: Audiovisual distractors do not interfere with participants performance in the working-memory task.** Participants' performance in remembering the sequence of fixation point color changes was high (group mean d′ = 2.99, SD = 0.91). Importantly, their sensitivity and criterion parameters did not vary as a function of gratings levels of contrast (d′ $F_{2,48} = 0.98$, $P = 0.37$, BF = 0.14 and c $F_{2,48} = 0.29$, $P = 0.75$, BF = 0.08) or auditory stimulus presentation (d′ $F_{1,24} = 0.308$, $P = 0.58$, BF = 0.20 and c $F_{1,24} = 0.44$, $P = 0.51$, BF = 0.21; Fig. 7). This demonstrates that the task-irrelevant audiovisual stimuli could be successfully ignored by the participants as they did not interfere with the color working-memory task. Likewise, RTs analyses did not show any difference as a function of grating contrast ($F_{2,48} = 0.9$, $P = 0.41$, BF = 0.12) or auditory presentation ($F_{1,24} = 0.01$, $P = 0.89$, BF = 0.17. Group mean RT = 880 ms, SD = 0.15).

**Experiment 2: Sounds do not enhance the maintenance of task-irrelevant visual information.** We found that the decoders trained and tested in the low contrast condition could not discriminate between S+ and S− trials (Supplementary Figs. 8, 9). This result indicates that diverting participants attention away from the visual gratings diminished the contrast response gain[65] compared to the first experiment, leading to a null decoding sensitivity for threshold-level visual stimuli. Conversely the neural activity driven by the high contrast stimuli was strong enough to be discriminated above chance level at multiple time points

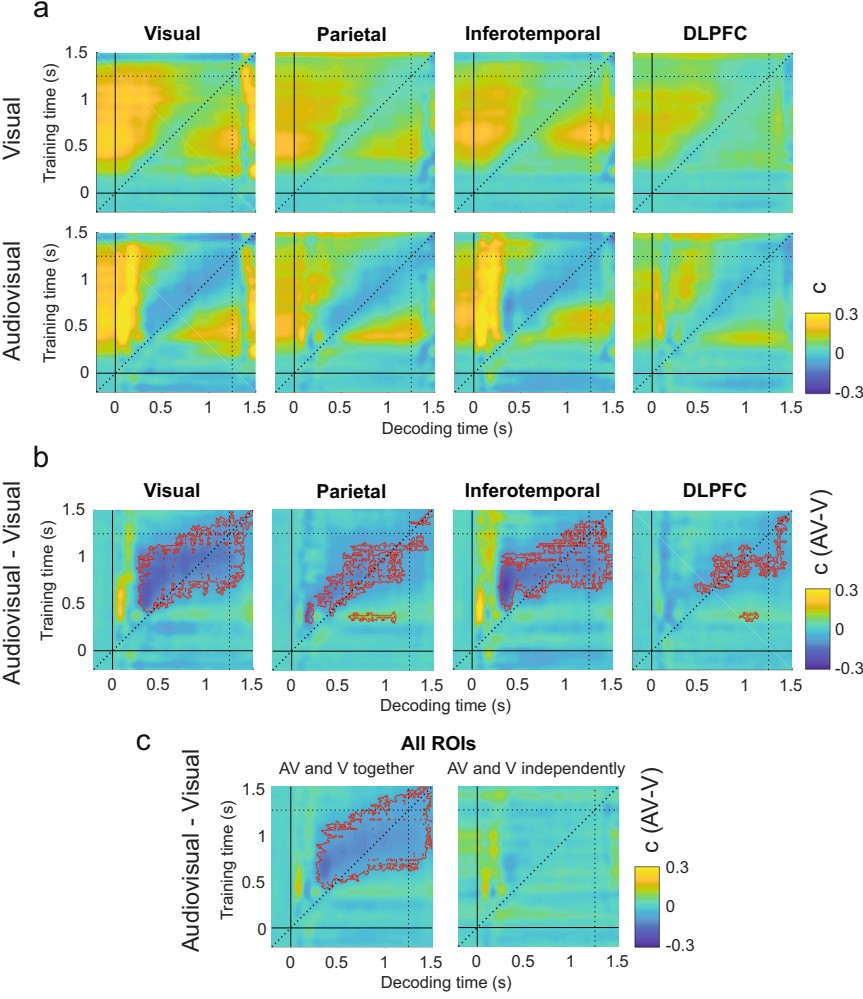

**Fig. 6 Visual criterion decoded from visual and audiovisual stimuli across ROIs.** Temporal generalization matrixes depict (**a**) the neural decoder's criterion parameter (**c**) in the visual (upper TGMs) and audiovisual (bottom TGMs) conditions and their difference (**b**) in each ROI. **c** Represents the differential decoded criterion averaged over ROIs with classifiers trained in the visual and audiovisual conditions together (left panel) or separately (right panel). Independently trained classifiers fail to capture the existing differences in criterion between the audiovisual and visual conditions. Red contours delimit significant clusters. Vertical and horizontal dotted lines represent the response phase onset.

(Fig. 8). Given the null sensitivity of the decoders in classifying the neural activity patterns associated to low contrast stimuli, we constrained our subsequent analyses to the high contrast condition.

We found that high contrast stimuli could be decoded as early as 100 ms in the visual cortex. Decoding accuracy peaked at 180 ms in the visual, inferotemporal and parietal ROIs ($P < 0.001$, $t_{max} = 7.4$, $t_{clust} = 3258$; $P < 0.001$, $t_{max} = 7.8$, $t_{clust} = 1312$; $P < 0.001$, $t_{max} = 6.1$, $t_{clust} = 801$) and around 350 ms in the dorsolateral prefrontal ROI ($P < 0.2$, $t_{max} = 5.29$, $t_{clust} = 23.5$). The d′ TGM profile in the high contrast condition showed that most of the significant clusters were distributed along the diagonal axis with weak information generalization (Fig. 8a). This is consistent with the highly dynamical information broadcasting that takes place during sensory processing[43,59], and suggest that the stimuli were conveniently ignored by the participants as they failed to trigger the sustained generalization pattern typically associated with visual awareness[43].

To investigate whether sounds enhanced the neural decodability of the unattended visual targets, we contrasted the audiovisual and visual d′ TGMs but we could not find significant differences between both modality conditions in any of the selected ROIs (Fig. 8b). Since this contrast yielded clear differences in our first experiment, but we observe a null result here when attention is directed away from the audiovisual stimuli, we conclude that the sound-induced enhancement in the maintenance of visual information is highly dependent on stimulus relevance, and it is therefore likely mediated by an endogenously controlled attentional mechanism.

**Experiment 2: Sounds drive visual activity patterns in visual cortex in a bottom-up fashion.** To test whether sounds influenced the neural decoder's criterion in the high contrast condition (Fig. 9a), we contrasted the audiovisual and visual c TGMs. We found a significant cluster ($P < 0.01$, $t_{min} = -2.07$, $t_{clust} = -105$) in the visual cortex ROI, meaning that decoders trained from 450 to 650 ms classified the activity patterns from 250 to 450 ms as S+ more often in the audiovisual compared to the visual trials (Fig. 9b). That is, sounds increased the proportion of neurally decoded hits but also of false alarms at around 350 ms (Supplementary Fig. 10). This result demonstrates that sounds can automatically evoke patterns of activity in the visual cortex that are often misclassified by a decoder as an actual visual stimulus.

One might argue that the sound-induced increase in false alarms reports might be unspecifically related to the decoder

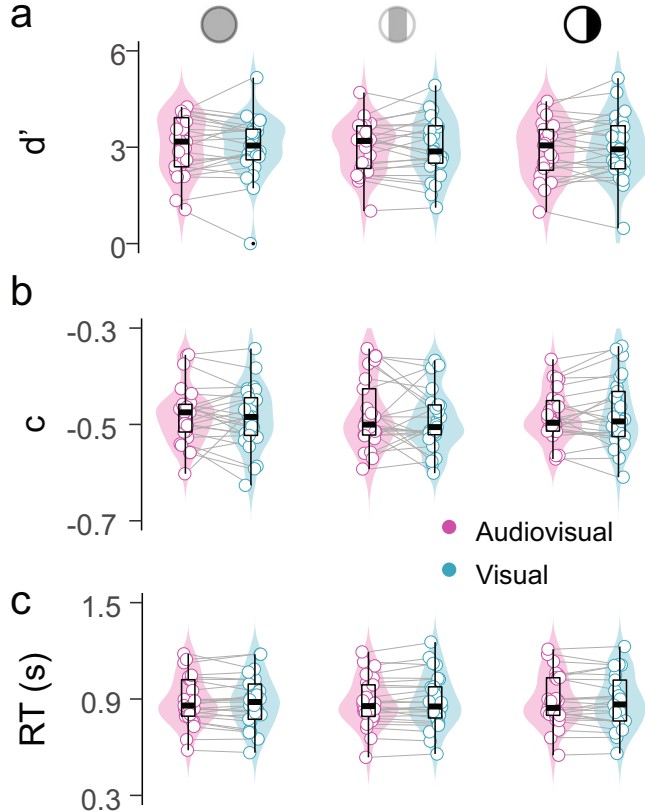

**Fig. 7 Behavioral performance in a working-memory task is not affected by the audiovisual distractors.** Behavioral results in the working-memory task: Sensitivity (**a**), criterion (**b**) and reaction times (**c**) are represented for each modality (audiovisual/visual), distractor contrast (S−/S+/S++) and participant. Inside the violin plot, the horizontal black line reflects the median, the thick box indicates quartiles, and whiskers 2 times the interquartile range. Gray horizontal lines connect participants scores across conditions.

picking up on univariate auditory ERF deflections present in both S+ and S− trials. This explanation is unlikely because the decoders that showed an increase in decoded false alarms were trained at a temporal window (450 to 650 ms) in which the neural activity in S− and S+ trials did not differ univariately (Supplementary Fig. 11). Moreover, the activity that was misclassified as S+ spanned along the second half of the auditory ERF but did not generalize to the first half even though both halves were almost symmetrical.

Our results are congruent with previous literature in showing that sounds activate early visual regions in a bottom-up fashion[6,7,31,39], and suggest that these activations encode stimulus-specific neural representations.

## Discussion

Visual detection is one of the most basic but fundamental perceptual processes carried out by the nervous system. Behind the apparent simplicity of detecting a stimulus, there is a complex and intermingled sequence of neural processes that bring an external sensory event into awareness. Multisensory research has demonstrated that visual detection is not impervious to stimulation in other sensory modalities. For instance, sounds can improve an observer's sensitivity while concurrently biasing their response criterion[11,13,14]. Therefore, an open question in visual neuroscience is how sounds modulate perceptual decisions during visual detection. Here, we addressed this question by analysing

source-reconstructed neural activity in different brain regions along the perceptual hierarchy involved in visual awareness[43,51]. In two different experiments, we used a multivariate decoding approach to elucidate the neural mechanisms by which sounds modulate observers' sensitivity and criterion in a visual detection task.

We found that sounds enhanced the maintenance of visual information over time. This effect of sounds on visual sensitivity can be characterized by different dynamical processing models (see model's description in Fig. 10) that vary in terms of their overall architecture (i.e., the number and the order of the processing stages) and whether they postulate that the sound-induced enhancement correlates with an increase of the amplitude, or the duration of a given processing stage[43,66].

Our results are better explained by a late maintenance model in which stimulus information, after being encoded in early sensory regions is remapped and stored as a decision variable (DV) until the response phase. According to this model's predictions, sounds would improve visual sensitivity by enhancing the maintenance over time of visual information encoded at 500 ms after the stimulus onset. The sound-induced enhancement of the information encoded at these specific time points does not seem like a coincidence, as 500 ms neural activity appears important for the perceptual decision itself (Fig. 3c).

Based on the temporal ordering of the significant clusters across ROIs (Supplementary Fig. 5) and in line with the currently accepted view that multisensory interactions depend upon a widely distributed network of brain regions[4,60,61,67], we propose that a late maintenance dynamical mechanism could be implemented, first by the encoding of the visual stimulus in the visual cortex. Concurrently, the encoded sensory information would be accumulated into a latent DV in parietal regions[46,47,55]. This is a stochastic process that, as demonstrated by the centroparietal activity analyses, can be optimized by a synchronous auditory stimulus. Interestingly, we found that in the audiovisual conditions, the centroparietal activity is drastically reduced at around 350 ms, suggesting that the ongoing process of evidence accumulation is reset. Such reset might correspond to a strategic reorienting of processing resources towards those sensory samples that more likely encode the visual target, that indeed manifests subsequently by a strong increase in evidence accumulation. On average, the evidence accumulation process reaches its peak at around 500 ms, and the DV is stored in the inferotemporal and DLPF cortices to protect them from interference with new incoming sensory input. Indeed, the interplay between these two brain regions has been demonstrated to play an important role in short-term memory during perceptual decisions[49,51,68–70]. As the response phase approaches, the participants can strategically reorient their feature-based attention towards the stored relevant stimulus information. This manifests by an enhanced reactivation of the target at 1000 ms in DLPFC[71,72], that subsequently leads to an enhanced reinstatement of the stimulus neural representations in visual cortex[72–74] coinciding with the response phase (at 1250 ms).

In a second experiment we found that when participants ignored the audiovisual stimuli, we could not replicate the same pattern of crossmodal interactions. This result reinforces the hypothesis that in the first experiment, sounds helped participants to guide their temporal attention in a top-down fashion, improving the processing and maintenance of task-relevant perceptual information[13,64]. We speculate that the sound-induced visual sensitivity enhancement could be partly mediated by a mechanism like retroception[63,75]. In retroception a post-stimulus visual cue can retrospectively amplify the signal gain of a target stored in perceptual memory that would otherwise have escaped consciousness. In the sound-induced visual enhancement such signal gain amplification would be instantiated by

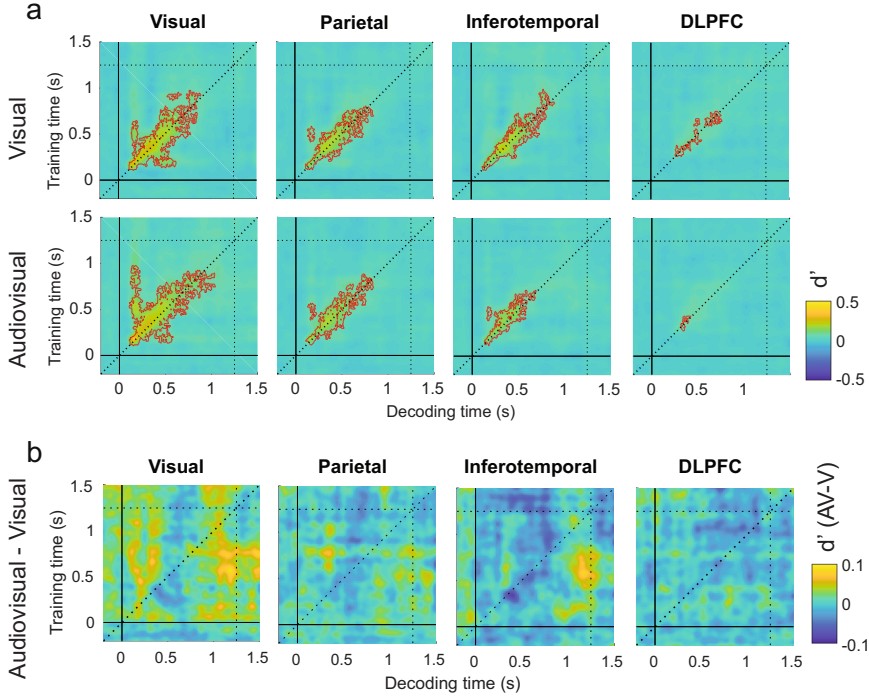

**Fig. 8 Visual sensitivity decoded from visual and audiovisual task-irrelevant stimuli across ROIs.** Temporal generalization matrixes illustrate (**a**) the decoded sensitivity parameters (d′) for the high contrast condition (S− vs. S++) in the visual (upper TGMs) and audiovisual (bottom TGMs) conditions and their difference (**b**) across ROIs. Red contours delimit significant clusters. Vertical and horizontal dotted lines represent the response phase onset.

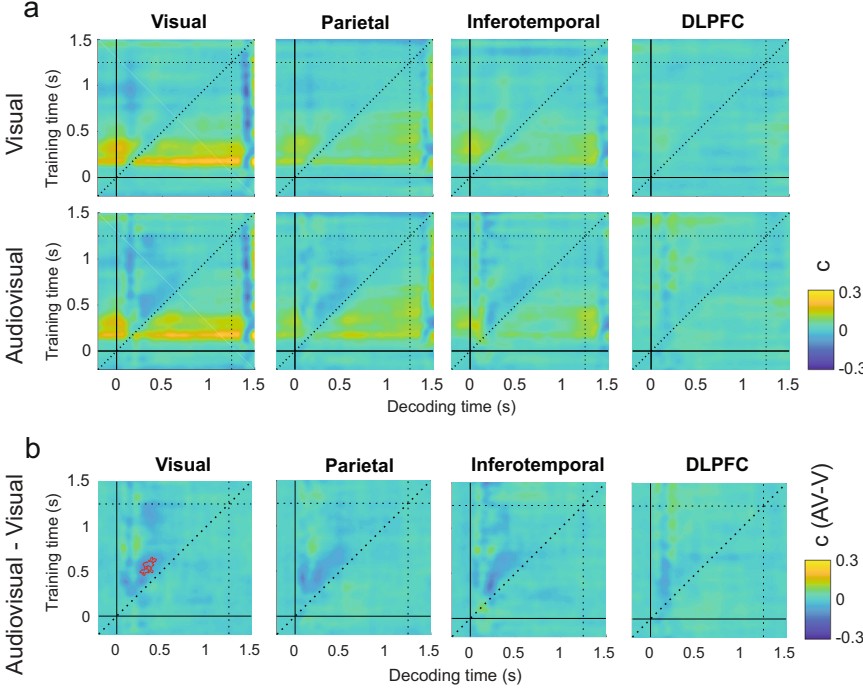

**Fig. 9 Visual criterion decoded from visual and audiovisual task-irrelevant stimuli across ROIs.** Temporal generalization matrixes illustrate (**a**) the decoded criterion parameters (d′) for the high contrast condition (S− vs. S++) in the visual (upper TGMs) and audiovisual (bottom TGMs) conditions and their difference (**b**) across ROIs. Red contours delimit significant clusters. Vertical and horizontal dotted lines represent the response phase onset.

an auditory cue instead, improving the accumulation and maintenance of perceptual information at post-sensory (i.e., decision) processing stages.

Although our results suggest that the crossmodal visual enhancement takes place through a late top-down controlled mechanism, this does not preclude that other more automatic and sensory-level multisensory mechanisms could also contribute to enhance visual detection sensitivity (although these might have escaped the sensitivity of our decoding analyses). For instance, audiovisual inputs might be integrated at subcortical level in the superior colliculus or through direct connections between early sensory areas[5–7,18,73,76]. Alternative accounts propose that sounds

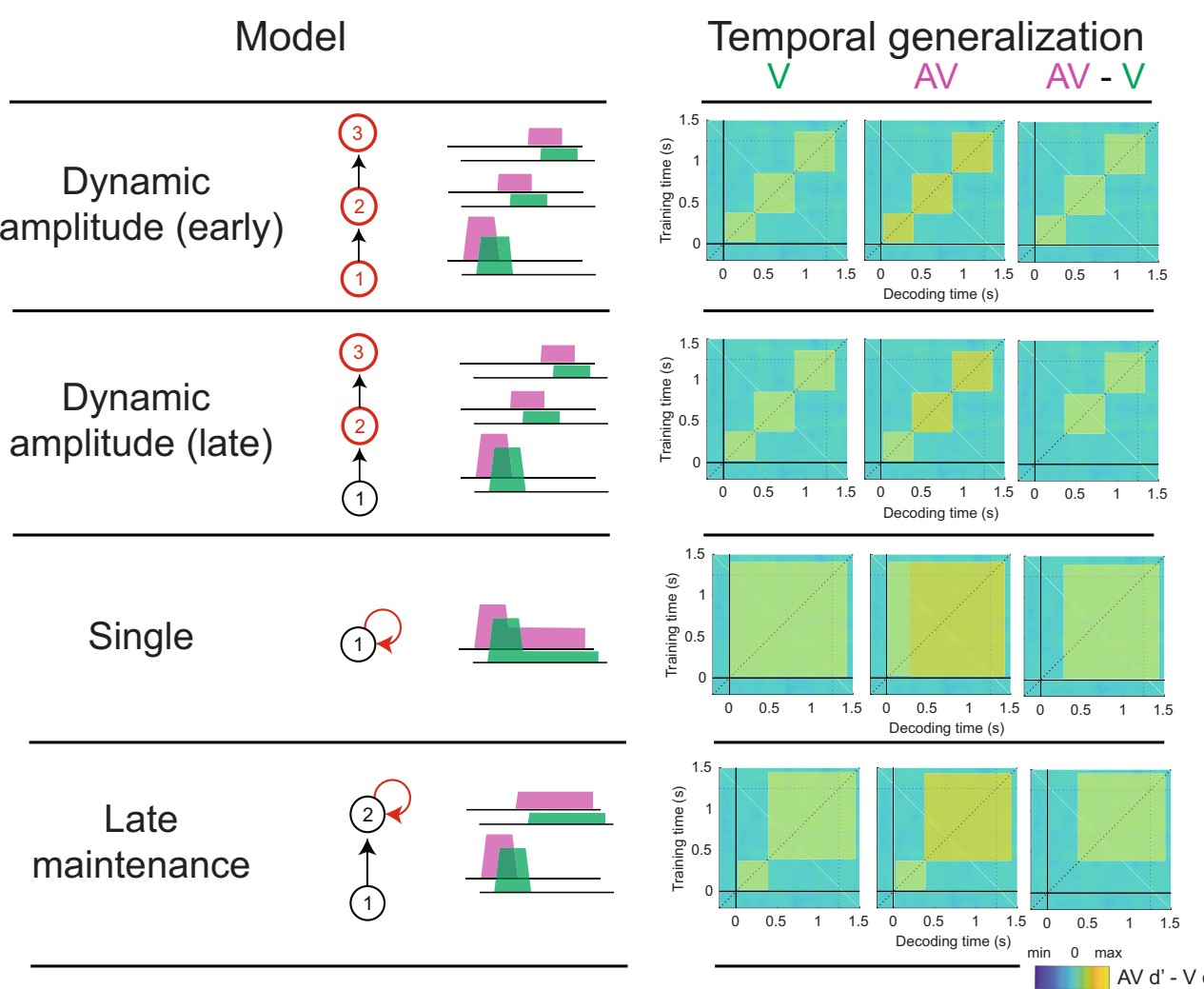

**Fig. 10 Candidate dynamical processing models beyond the sound-induced visual detection enhancement.** Sounds might improve the detectability of a visual signal through different mechanisms. To characterize the dynamic nature of information processing we describe four models that vary in terms of the number and ordering of processing stages. We propose that sounds could enhance detection by increasing (1) the decoding amplitude or (2) the duration of one or several processing stages: The dynamic amplitude models broadcast target information across a sequence of short-lived processing stages whose amplitude codes for its detectability. We hypothesize that sounds might act upon one of these stages, boosting the decoding amplitude at early or late processing stages. The single stage model encodes and maintains the target information within the same processing stage. We propose that sounds might improve the maintenance of information within the initial processing stage. The late maintenance model transmits the encoded information to a later processing stage where task-relevant information is sustained through recurrent processing loops. Sounds might enhance the maintenance of information within the later processing stage. Vertical arrows represent information transfer from one neural stage to the next. Circular arrows represent recurrent feedback loops that maintain information over time. The key process that characterizes the effect of sounds in each model is highlighted in red.

could exert a modulatory influence by resetting the phase of ongoing oscillations in visual cortex, such that co-occurring visual targets align with high-excitability ideal phases[19,74,77]. These two sensory-level mechanisms of audiovisual interplay have been well-described at an electrophysiological level, however future studies must continue elucidating which are their specific functional role in multisensory perception.

In this work, in addition to the sound-induced enhancement in visual sensitivity, we showed that sounds reduced the participants criterion in visual detection. Consistently with this result, in our first experiment the decoding analyses showed that sounds biased the decoders to classify the audiovisual trials as S+ more often than the visual ones. The classical interpretation of this decoding bias would imply that participants, after hearing the sound, are more inclined to believe that the target has been presented (i.e., decision-level bias). However, in our second experiment in which participants did not have to make any decision about the presence

or absence of the visual targets, we found again that sounds automatically evoked patterns of activity in early visual regions similar to the patterns evoked by an actual visual stimulus. This result was characterized by a sound-induced increase in the proportion of decoded false alarms. We speculate that sound-induced activity in visual regions could be subjectively mis-interpreted by the observer as an actual visual stimulus, leading to perceptual-level biases. Thus, the decoding criterion biases reported in the first experiment might be partially explained by decision and perceptual-level biases together. Indeed, there is solid empirical evidence proving that sounds can induce visual illusions. For instance, in the double-flash illusion presenting a single visual stimulus paired with two auditory stimuli induces the percept of an additional illusory visual stimulus with identical physical properties as the inducer[78–80].

Therefore, our current results are consistent with previous work[6,30,31,81] by demonstrating that sounds alone can modulate

the activity in early visual areas in a bottom-up fashion. It is important to note that whereas we replicated that sounds induced univariate activity modulations in most of our selected ROIs and specially in the visual cortex (Supplementary Fig. 12), we must be cautious in interpreting them as genuine crossmodal sensory modulations. Despite the spatial filtering applied to the ERF signal using beamforming source reconstruction, auditory activity can potentially spread from the auditory cortex to the virtual channels in adjacent brain regions, including the visual cortex. Thus, the use of multivariate decoding is more accurate in discriminating visual-stimulus-specific activity patterns from auditory trials regardless overall changes on univariate activity.

Previous research has also shown that complex natural sounds evoke category-specific information that can be read-out from early visual regions[82]. Similarly, it has been demonstrated that auditory motion can activate higher order visual regions like occipito-temporal cortex (V5/hMT+)[83,84]. All these findings together highlight that audiovisual interactions are pervasive in the human brain. Nevertheless, the neural mechanisms by which simple and complex auditory stimuli are fed back to striate and extrastriate visual areas might be qualitatively different and further research is required to better characterize the different mechanisms supporting audiovisual interactions. In brief, our results provide neural evidence for an alternative hypothesis on the interpretation of SDT parameters[23,24] that challenges the classical assumption postulating that sound-induced reductions in criterion necessarily correspond to decision-level biases.

Retrograde tracing studies in monkeys have revealed the existence of direct connections between the auditory the visual cortex. These connections are heterogeneously distributed across the visual field with more peripheral eccentricities receiving denser projections[57,76]. Given that such asymmetrical connectivity pattern could determine the mechanism by which sounds interact with visual detection, in our first experiment we tested the effect of sounds on visual stimuli presented at central and peripheral visual eccentricities. Our behavioral and neural decoding analyses in the first experiment yielded converging results at both eccentricity conditions, allowing us to conclude that in this study, the mechanisms by which sounds impact visual detection do not depend on retinal eccentricity. Future studies should further investigate whether the heterogenous auditory-to-visual cortex connectivity pattern reported in monkeys also exists in humans[85]. If it does, the next question to answer is whether it plays a functional role for multisensory processing, as proposed in previous studies[86,87].

In recent years it has been demonstrated that the visual and auditory systems are heavily interconnected at multiple processing levels[4,8,9]. Although the effect of sounds on visual detection has been well described at a behavioral level, it remained unclear which specific neural mechanisms give support to these crossmodal interactions. In this study we implemented a multivariate decoding approach to neurally dissociate the contribution of sounds to sensitivity and criterion modulations in a visual detection task. Our results demonstrate that multisensory interplay in visual detection does not exclusively rely on sensory-level crossmodal interaction. Instead, it unfolds at multiple levels of the perceptual hierarchy improving the amplitude of the encoded visual representations and their temporal stability. In addition, our results also help to reconcile two opposing views by showing that audiovisual interactions in detection rely on parallel top-down and bottom-up crossmodal mechanisms. Whereas the sound-induced improvement in visual sensitivity is mediated through a widely distributed network of brain regions in which the accumulation and maintenance of post-sensory visual information is improved via top-down mechanism, sound-induced reductions in criterion do not exclusively reflect decision-level

biases. Instead, these biases can also be induced throughout bottom-up auditory dependent modulations of visual cortex activity. In the future, the combination of TG decoding with more specific functional and anatomical localizers will help to understand with a greater level of detail how the different brain regions collaborate to enhance visual detection when an auditory stimulus is presented, and how specifically visual information is transformed across the multiple stages that conform the perceptual decision-making hierarchy.

## Methods

**Subjects**. Twenty-five healthy human volunteers with normal or corrected-to-normal vision and audition participated in the first (17 females, mean age = 24 years, SD = 6 years) and second experiment (12 females, mean age = 25 years, SD = 7 years). One subject in the first and one subject in the second experiment were excluded during the preprocessing due to insufficient data quality (severe eye and muscle artifacts and poor performance). Participants received either monetary compensation or study credits. The study was approved by the local ethics committee (CMO Arnhem-Nijmegen, Radboud University Medical Center) under the general ethics approval ("Imaging Human Cognition", CMO 2014/288), and the experiment was conducted in compliance with these guidelines. Written informed consent was obtained from each individual prior to the beginning of the experiment.

**Stimuli**. Visual stimuli were back-projected onto a plexiglass screen using a PROPixx projector at 120 Hz. In the first experiment the screen region where the targets could appear was delimited at the beginning of each trial using parafoveal (center condition; inner and outer perimeters 1° and 5.5° radius) or perifoveal annular (periphery condition; inner and outer perimeters 5.5° and 11° radius respectively) noise patches centered on the fixation point (Fig. 1b). In the second experiment all the stimuli were presented parafoveally (center). The noise patches were created by smoothing pixel-by-pixel Gaussian noise through a 2D Gaussian smoothing filter[88]. Signal-present (S+) stimuli consisted of vertical sinusoidal gratings (spatial frequency of 0.5 cycles/° and random phase sampled from a uniform distribution) added to the previously generated noise patches. Thereby, the noise structure of the placeholder and signal stimuli was the same within trials, but it was randomly generated for each new trial. The fixation point was a circle (radius = 1°) presented at the center of screen. In the first experiment the fixation point color was black and in the second experiment light gray (luminance: 405 cd/m²). All the visual stimuli were displayed on a gray background (50% of maximum pixel intensity, luminance: 321 cd/m²). Auditory stimuli were presented through in-ear air conducting MEG compatible headphones and consisted of binaurally delivered pure tones at 1000 Hz (70 dB, 5 ms rise/fall to avoid clicks; 16 bit mono; 44.100 Hz digitization). All the stimuli were generated and presented using MATLAB 15 (The Mathworks, Inc., Natick, Massachusetts, United States) and the Psychophysics Toolbox extensions[89].

**Procedure and experimental design**. In the first experiment participants performed a visual detection task (Fig. 1a). First, the participants completed a 5 min behavioral training session in which they were familiarized with the new environment and task. After the practice session, we used an adaptive staircase (Quest; Watson & Pelli, 1983)[90] to estimate the level of contrast at which each subject correctly detected a vertical grating in 70% of the cases. The trial sequence used during the Quest procedure was similar to the one used during the main task detection blocks (see below) except that subjects were exclusively presented with the visual condition. We used two independent staircases to estimate the contrast for the center and periphery conditions and only the trials in which the grating was presented were used to update the staircase. Once we estimated the participant threshold, participants started the main task blocks. The appearance of a central fixation point signaled the beginning of each trial. This fixation point was kept on the screen for a variable period of time of 750 to 1000 ms (randomly drawn from a uniform distribution) and determined the inter-trial-interval (ITI). Then, a circular or an annular noise placeholder was displayed on the screen. The noise placeholder constrained the visual space prior to the presentation of the visual target indicating at which visual eccentricity the target might occur (center or periphery, both conditions were equiprobable) and eliminating any possible spatial uncertainty. After a random period of time between 1000 to 1500 ms, the target grating, referred to as signal-present trial (S+) or the same noise placeholder, referred to as signal-absent trial (S−) was displayed for 33 ms (4 frames). Importantly, in half of the trials an auditory tone of the same duration as the target was presented in synchrony with the [S+|S−] event onset. Subsequently, the noise placeholder alone was displayed again and remained on the screen for a fixed period of 1250 ms. Then, the letters 'Y' and 'N' (as abbreviations for Yes and No, respectively) were centered around 4° the fixation dot. Subjects reported their decision as to whether or not they had detected a vertical grating by pressing a button with either the left or the right-hand thumb, corresponding to the position of the letter that matched their decision. The position of the letters ('Y' left and 'N' right, or 'N' left and 'Y'

right) was randomized across trials to orthogonalize perceptual decision and motor response preparation. Finally, after a response period of 2000 ms the fixation point turned green, red or white for 250 ms signaling correct, incorrect or non-registered responses respectively. Participants performed a total of 576 trials divided in 6 blocks of 10 min. There were a total of 8 different experimental conditions: stimulus type (signal-present/signal-absent) × modality (visual/audiovisual) × eccentricity (center/periphery), and participants ran a total of 72 trials per combination of conditions. During the main task the grating contrast values were kept fixed within blocks. If at the end of one block, due to learning or tiredness participants showed near perfect (>90% correct responses) or chance level (<55% correct responses) performance, the contrast grating was updated using the new threshold value estimated by the Quest procedure. This contrast adjustment happened in 33% of blocks.

In the second experiment we designed a task that forced participants to ignore the previously task-relevant visual gratings and sounds, and test whether the effects found in the first experiment were simply due to participants being involved in a visual detection task. Participants completed a 5 min behavioral training session in which they were familiarized with the task (see below). After the practice session, as in the first experiment we used a Quest to estimate the level of contrast at which each subject correctly detected a vertical grating in 70% of the cases. Then, after ensuring that participants had understood the main task, we continued with the experiment. In the main task blocks the stimulus sequence was identical to the one used in the first experiment with some minor but relevant modifications (Fig. 1c). Here, although the vertical gratings and sounds were presented in each trial as in the first experiment, the participant's task was to ignore them and perform a working-memory task on the fixation point: The fixation point changed to magenta, yellow or cyan briefly for 33 ms once before and once after the ignored [S+|S−] event and participants had to memorize these two color changes. At the end of the trial and during the response phase, the fixation point changed a third time revealing a color that could be the same or different (in 50% of the trials) as one of the previously memorized colors. Participants had to choose whether this third color was repeated or not by selecting between 'S' or 'D' (as abbreviations for Same and Different respectively) using the left or right thumb on the button box. The fixation point color sequences were randomly generated in each trial by selecting without repetition two of the three previously described colors. To ensure that participants maintained their attention steadily on the fixation point during the whole trial, the first two fixation point color changes happened during the presentation of the first and second noise placeholders, and their onset varied randomly from trial to trial 300 to 750 ms relative to the onset of the S+ or S− events. In order to avoid strong luminance variations at the fixation point during color changes, by default the fixation point was colored in light gray at a luminance value near to the average luminance of the color changes. We set the duration of the fixation point presentation (ITI = 500 to 750 ms), the first noise placeholder (600 to 1500 ms) and the response time (1500 ms). Participants performed a total of 648 trials divided in 6 blocks of 10 min. There were a total of 6 experimental conditions of interest: stimulus type (S−|S+|S++) × modality (visual/audiovisual), and at the end of the experiment each participant underwent a total of 108 trials per combination of conditions. The contrast level used for the gratings used in the high contrast S++ condition was generated by doubling the contrast estimated for the S+ condition. This condition served to ensure above chance-level decoding classification of signal-present vs signal-absent conditions and to test whether the audiovisual interaction strength changed as a function of the stimulus bottom-up saliency. The vertical grating stimuli where only presented at one eccentricity (center). Since it was impossible to assess whether the subjective visibility of the gratings changed across blocks, as participants did not make decisions about the gratings absence or presence, the contrast value was kept fixed during the whole experiment.

**Behavioral data analyses**. In the first experiment, we quantified participants sensitivity (d') to discriminate S+ from S− trials using the SDT (yes-no paradigm). In addition, to estimate participants biases to report S+ or S− independently from their visual sensitivity, we computed the criterion (c) parameter[91]. SDT sensitivity and criterion parameters were calculated for each condition and block by coding S+ trials correctly reported as 'signal' as hits and S- trials incorrectly reported as 'signal' as false alarms. This way, lower c parameter values represented a larger bias to report the signal as present. In the second experiment the participants task was to remember a sequence of items and report whether the third item was the same or different from the previously presented items. Therefore, we again applied SDT to calculate participants sensitivity. However, given that participants had to compare whether the fixation point colors were same or different, we modeled the SDT parameters assuming a same-different decision paradigm[91]. Here, high d' values represented better performance in discriminating whether the last fixation point color had been presented or not during the trial sequence. Complementarily, the c parameter indexed the participant's bias to report the last fixation point color as repeated (same) or different.

**MEG recording and preprocessing**. Whole-brain neural recordings were registered using a 275-channel MEG system with axial gradiometers (CTF MEG Systems, VSM MedTech Ltd.) located in a magnetically shielded room. Participants' eye-movements and blinks were tracked online using an EyeLink 1000 (SR

Research). Throughout the experiment, head position was monitored online and corrected if necessary using three fiducial coils that were placed on the nasion and on earplugs in both ears. If subjects moved their head more than 5 mm from the starting position, they were repositioned during block breaks. All signals were sampled at a rate of 1200 Hz. Data preprocessing was carried out offline using FieldTrip (www.fieldtriptoolbox.org). The data were epoched from 2000 ms before and 1500 ms after the signal-present/absent event onset. To identify artifacts, the variance (collapsed over channels and time) was calculated for each trial. Those trials and channels with large variances were subsequently selected for manual inspection and removed if they contained excessive and irregular artifacts. Additionally, trials without participant's response or containing eye blinks within the interval of 100 ms before or after the target presentation were removed from subsequent analyses. We used independent component analysis to remove regular artifacts, such as heartbeats and eye blinks[92]. For each subject, the independent components were inspected manually before removal. "Bad" channels showing SQUID jumps or other artifacts were interpolated to the weighted mean of neighboring channels. Weights correspond to the physical distance between sensors (ft_repairchannel, as implemented in FieldTrip[93]). For the main analyses, data were low-pass filtered using a Butterworth filter with a frequency cutoff of 30 Hz and subsequently downsampled to 100 Hz. No detrending was applied for any analysis. Finally, the data were baseline corrected on the interval of −200 ms to the [S+|S−] event onset (0 ms).

**Source reconstruction and ROI generation**. For each participant we build volume conduction models based on single-shell model of the standard Montreal Neurological Institute (MNI) anatomical atlas[94]. Then, we used them to construct search grids (10-mm resolution). For each grid point, lead fields were computed with a reduced rank, which removes the sensitivity to the direction perpendicular to the external boundary of the volume conduction model.

In order to gain insight on how the auditory stimuli modulate the processing of visual stimuli across the perceptual hierarchy, we generated four neuroanatomically defined regions of interest (ROIs; Fig. 3a) using the AAL anatomical atlas: These were the visual cortex, the parietal cortex, the inferotemporal cortex and the dorsolateral prefrontal cortex. These brain regions have been typically associated with perceptual decisions in visual detection tasks (see Supplementary Table 1). Using the (inverse) covariance matrix computed from the combined visual and audiovisual trials (−0.2 to 1.5 s, time-locked to the target onset; 10% normalization), the volume conduction model, and the lead field, we applied a linearly constrained minimum variance[95] beamformer approach in Fieldtrip[93] to build a common spatial filter for each grid point and participant. Finally, to spatially constrain the analyses to the previously selected neuroanatomical regions, we projected the sensor-level ERFs timeseries through those virtual channels that spatially overlapped with our ROIs.

**Multivariate decoding analysis**. We applied a multivariate pattern analysis approach to classify single trials as S+ vs S− as a function of the neural activity measured from the virtual channels composing each ROI from −0.2 to 1.5 s. The method that we applied was largely based on linear discriminant analysis[59,96]. First, in order to minimize time point by time point absolute univariate differences between the visual and audiovisual conditions, we z-scored the ERF activity across virtual channels for each time point and considering the visual and audiovisual conditions independently. Then, using the patterns of activity measured at the multiple virtual channels (i.e., generally termed as features) contained in one of our ROIs, we calculated the weights vector w that optimally discriminates between S+ and S− trials (Eq. 1).

$$w = \widetilde{\Sigma}_c^{-1} (\hat{\mu}_2 - \hat{\mu}_1) \tag{1}$$

$\hat{\mu}_1$ and $\hat{\mu}_2$ are two column vectors of length F (i.e., number of features) for a given time point, representing the neural activity averaged across the S+ ($\hat{\mu}_2$) and S− ($\hat{\mu}_1$) trials respectively, and $\widetilde{\Sigma}_c^{-1}$ is the common regularized covariance matrix. The regularization parameter was optimized in preliminary tests using cross-validation and was kept fixed for all subsequent analyses. To make the encoding weights comparable across time points, we added a normalization factor (denominator in Eq. 2) to the weights vector such that the mean difference in the decoded signal between classes equals a value of one.

$$w = \frac{\widetilde{\Sigma}_c^{-1} (\hat{\mu}_2 - \hat{\mu}_1)}{(\hat{\mu}_2 - \hat{\mu}_1)^T \widetilde{\Sigma}_c^{-1} (\hat{\mu}_2 - \hat{\mu}_1)} \tag{2}$$

Next, to assess whether the learned weights could discriminate between S+ and S− trials we cross-validated our decoder (Eq. 3) in a different set of trials X.

$$y = w^T X \tag{3}$$

X is a matrix of size F × N, where N represents the number of trials and F the number of features present in the independent dataset. By multiplying X by the weights transpose matrix ($w^T$) that we previously estimated from the training dataset (Eq. 2), we obtained the decoder output, termed here as the discriminant channel. If there is information in the neural signal pertaining to the classes to be decoded, we expect the mean discriminant channel amplitude to be $\bar{y}_2 > \bar{y}_1$,

whereas if no information is available in the neural signals, we must find $\bar{y}_1 = \bar{y}_2$. In order to classify a trial as S+ or S−, we set a cutoff value. If the difference between the discriminant channels $\bar{y}_2$ and $\bar{y}_1$ was larger than 0, the trial was classified as S+. Instead, if the difference between the discriminant channels $\bar{y}_2$ and $\bar{y}_1$ was smaller than 0, the trial was classified as S−. To avoid double dipping[97] we adopted a leave-one-out cross-validation approach where we randomly divided the trials of the dataset in five evenly distributed folds. We built an unbiased classifier by training our discriminant model in four of the five folds that contained visual and audio-visual trials. Subsequently, we tested the classifier in the remaining fold. We repeated the same process until all the folds were used as training and test sets. Moreover, given potentially unequal trial numbers for each visual and audiovisual condition, we repeated the same process 50-times and averaged the final discriminant channel output for each trial.

We calculated sensitivity and criterion parameters from neural activity. For instance, if the classifier categorized a S+ trial as S+, we coded the trial as a hit. Instead, if the classifier wrongly categorized a S- trial as S+, we coded the trial as a false alarm (Fig. 3b). Using the same approach as in a signal detection theory Yes-No paradigm[10], we estimated sensitivity (d′) and criterion (c) parameters for each time point in each trial based on neural activity derived hits and false alarms. The d' parameter allowed us to evaluate the sensitivity of the classifier to discriminate between S+ and S− trials in the visual and audiovisual conditions. Time points with d′ > 0 represents above chance-level visual stimuli classification. On the other hand, the c parameter indexed the bias of the classifier to classify the trials as S+ or S− in the visual and audiovisual conditions. Specifically, time points with lower c values represent stronger biases to report the signal presence regardless the actual presence or absence of the signal. This procedure allowed us to directly compare participants behavioral performance based on signal detection theory parameters with the same parameters estimated from neural activity during the perceptual decision making.

The decoding analysis outlined above was implemented in a time-resolved manner by applying it sequentially at each time point in steps of 10 ms, resulting in a decoders array of the same length as the number of trial time points. To characterize the temporal organization of the neural processes that underlie the auditory contributions to visual processing (Fig. 2), we implemented a temporal generalization (TG) method[41]. Each decoder trained on any specific time point was applied to all other time points. If we average the decoder output over trials, this results in a squared temporal generalization matrix (TGM) with training time × decoding time values per condition. The diagonal values in the TGM contain the estimated parameters for the decoders trained and cross-validated in the same time points ($t_{train} = t_{test}$). Instead, the row values in the TGM represent how a specific decoder trained at a time point $t_{train}$ in both visual and audiovisual conditions classifies visual and audiovisual trials as signal-present or signal-absent trials at earlier and later time points. In addition, a column gives insight into whether the neural patterns of activation of the two conditions can be discriminated at time point $t_{test}$ on the basis of the decoders trained on all other time points. In summary, observing that the same decoder can separate between conditions at multiple time points give us relevant information about how persistent in time neural representations are. For illustration purposes, the data present in the TGMs have been smoothed using a 2D Gaussian kernel (SD = 3)

**Correlation between neurally decoded and behaviorally estimated SDT parameters.** We used the LDA classifier predictions to generate the TGMs with the d' and c parameters for each one of the six experimental blocks. To test whether the neurally decoded parameters were associated with the participants performance, we correlated (Pearson correlation) the six neurally decoded d's with the six behaviorally estimated d's in each TGM train-test temporal combination (Supplementary Fig. 13). The same correlation procedure was repeated on the criterion parameters. We ended having 24 correlation maps (one per participant) for each SDT parameter. Using a cluster-based permutation approach we tested in which decoding train-test combinations of the TGM the mean of the group-level distribution of correlation coefficients was significantly different from 0.

**Centroparietal activity univariate analyses.** EEG activity registered in centroparietal sensors is strongly correlated with changes in evidence accumulation and decision formation processes[55,98]. To investigate whether and how sounds affected evidence accumulation in visual detection, we averaged the planar-combined ERF activity measured over eight centroparietal MEG sensors (MZC03, MZC04, MLC55, MRC55, MLC63, MRC63} for the visual and audiovisual conditions. Next, we quantified sound-induced changes in evidence accumulation gain by calculating changes in centroparietal sensors activity as a function of time. To do that, we fitted a linear regression model for each time point using a 200 ms sliding window. Additional control analyses showed that the results were qualitatively invariant to the extent of the used sliding window.

**Statistics and reproducibility.** We recruited 24 participants per experiment. The sample size was determined prior to data collection, and ensured 80% power to detect medium-to-large effects (Cohen's d > 0.6). We end up recruiting 25 participants per experiment as we had to excluded two participants (one in each experiment) due to insufficient data quality.

To statistically assess in which training and decoding time combinations of the TGM the decoder successfully discriminated the trials as S+ or S− above chance level, we contrasted the d' values against 0. We applied cluster-based permutation tests[99]. This procedure controls for multiple comparisons across training and decoding time combinations by leveraging the inherently correlated nature of neighboring observations. TGM could be compared univariately against 0 using a t-score. Positive and negative clusters were then formed separately by grouping temporally adjacent data points whose corresponding P-values were lower than 0.05 (two-tailed). Cluster-level statistics were calculated using the 'weighted cluster mass' method, a statistic that combines cluster size and intensity[100]. To obtain a null distribution of this cluster-level statistic, we repeated the same process 1000 times permuting trial-condition associations in each iteration. This yields cluster p-values, corrected for multiple comparisons. The null hypothesis was rejected for those clusters for which the p-value were smaller than 0.05, compared to the null distribution of cluster-level statistics. Finally, in order to compare how sounds modulated d' and c at different time points, we also applied cluster-based permutation tests but contrasting the audiovisual and visual TGMs with each other. In the results section, for each independent analysis we report the $t_{max/min}$ statistics within the most positive/negative significant clusters and its corresponding cluster-level t-value.

**Reporting summary**. Further information on research design is available in the Nature Portfolio Reporting Summary linked to this article.

## Data availability

Source data underlying Figs. 2, 7 are available in Supplementary Data 1, 2, respectively. All data is available online at the Donders Repository at https://doi.org/10.34973/2m6r-4167.

## Code availability

All codes used to run the task, process data, and generate figures are available at the Donders Repository at https://doi.org/10.34973/2m6r-4167.

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

## Acknowledgements

We are very grateful to Pim Mostert and Valentin Wyart for sharing their code and initial advice on the MEG and behavioral analyses. We thank Salvador Soto-Faraco for helpful comments on an earlier draft of the manuscript. This work was supported by the Netherlands Organization for Scientific Research (NWO Veni grant 016.Veni.198.065 awarded to ES and Vidi grant 452-13-016 awarded to FPdL), the EC Horizon 2020 Program (ERC starting grant 678286 awarded to FPdL), the Spain Ministerio de Ciencia, Innovación y Universidades (JIN RTI2018-100977-J-I00 awarded to APB) and AGAUR Generalitat de Catalunya (BP-00213 awarded to APB).

## Author contributions

A.P.B., and F.L. conception and design of research; A.P.B. performed experiments; A.P.B. and E.S. analysed data; A.P.B. drafted paper; A.P.B., E.S., and F.L. edited, revised, and approved the final version of the paper.

## Competing interests

The authors declare no competing interests.
