## [Peer Review File · Communications Biology]

Reviewers' comments:

Reviewer #1 (Remarks to the Author):

This is a fascinating study on crossmodal interaction (audiovisual) using temporal generalization analysis of source-space MEG data. The authors first measure the sensitivity (d') and criterion (c) using behavioral analysis and then use Latent Discriminant Analysis (LDA) to derive the same from MEG data. The sensitivity indicates the ability to discriminate between two conditions while the criterion score reflects the threshold for detection. Lower criterion leads to more false alarms. The authors show that these parameters derived from behavior and experiment are correlated.

Next, two experiments are conducted to answer specific neuroscientific hypothesis. In the first experiment, the authors test the hypothesis that sounds enhance detection due to crossmodal interactions and increase the false alarm rate (lower criterion). In the second experiment, the authors test the hypothesis that crossmodal interactions are enhanced due to attention and whether or not decision-level bias plays a role in the enhancement. While I cannot comment on the audiovisual integration literature, I find the machine learning analysis that has been used to probe psychological measures innovative. Specific comments are noted below:

Major points

1. Given that the correlation between behavioral and decoded criterion parameters is extremely low (around 0.1; Fig 3D), how reliable are the conclusions in the other figures? It is also not clear how many values are these correlations computed over? Are they for 24 subjects? Ideally a scatter plot of behavioral vs decoded psychological scores with 24 points will clearly show how reliable the proxy measures are.
2. The time generalization matrices look suspiciously similar between different brain regions. Have the authors verified that this is not an artifact of the data processing pipeline? It might be helpful to show in a figure: a) the source localization on the cortex at the time instant that the evoked response peaks, and b) the virtual channel for one condition and subject in the four ROIs being analyzed. It will verify that the noise covariance was correctly regularized and noise was effectively suppressed when computing the source estimate.
3. Related to the above point, have the authors considered what the upper triangle (where differences are observed in Figure 4 for example) of the TGM means? It seems to indicate that neural processes later in time can explain neural processes earlier in time pointing to an acausal phenomena.
4. Is there a bias in the LDA classifier somewhere because the baseline period in Figure S4 does not show 0.5 as the hit rate?

Minor points

1. The inverse of the covariance matrix has been called the covariance matrix in the methods section
2. The use of tilde is not clarified. Notations need more clarity and consistency.
3. The quality of certain figures is low. They appear to be screenshots rather than high resolution images for pdf
4. Signal Detection Theory (SDT) abbreviation needs expansion before use
5. Authors refer to "folds" as "manifolds". Manifolds has a specific meaning in mathematics which is perhaps not what they mean in the context.
6. References for ICA and spatial interpolation are missing
7. Expand abbreviation SM to supplementary material
8. "surface of the volume conduction model" seems like a weird phrase

9. References for visual sensitivity and criterion parameter are not fully appropriate as none of the references actually explain what these parameters are or how they are computed. Alternatively, the authors should explain this in the methods section and refer to the methods section.

10. In several parts of the text, the authors refer to the supplementary material. In my opinion, the main text should be readable without supplementary figures and references to them should be sparingly used. It breaks the flow of the reading.

Reviewer #2 (Remarks to the Author):

Thanks for giving me the opportunity to review your manuscript entitled "Seeing sounds: Neural mechanisms underlying auditory contributions to visual detection". In this manuscript, Perez-Bellido et al. use two complementary audiovisual paradigms to show that an additional sound played alongside a visual grating can facilitate detection and the presence of the grating can be decoded from neural activity using a decoder trained on visual and audiovisual trials together. These enhancements in sensitivity come with a concomitant criterion shift whose origin appears to be dependent on top-down controlled attention, as this shift wasn't present in experiment two. The authors interpret their results as broad effects of the integration of audiovisual stimuli across the entire processing hierarchy in time and space (ROIs).

Overall, this investigation of the mechanisms underlying the neural integration of audiovisual signals, their point in time, and effects of top-down control is timely. Multiple groups have attempted to shed light on these questions in recent years, but important questions remain to be answered. The present manuscript benefits from combining a traditional visual detection task with simple auditory tones. This is different from some previous paradigms, which used only simple or rather complex stimuli in both modalities. The focus on signal detection theory for quantifying the characteristics of the neural decoder is elegant and less common in the field. The pirate plots of behavioural and signal detection results depict these data and their different elements well. Taken together, while I appreciate the nice design of the two complementary experiments, the incorporation of visual eccentricity, and the manuscript's well-written style, I think the present version would benefit from further evidence to deliver on its ambitious goals. To foreshadow my recommendations, the temporal and scalp map results should be explored further by linking them to a functional interpretation more strongly and the statistical analysis would benefit from some considerations to corroborate the authors' claims unequivocally.

Specific comments with recommendations:

Major concerns

Given that one part of the authors question was to investigate where potential enhancements of auditory sounds would take place in the brain (e.g., early effects due to connections between primary visual and auditory cortices), I was surprised to see that the ROIs did not include an additional ROI for primary auditory areas (superior temporal lobe). The results presented in figure S12 suggest to me that the information encoded at the peak time point 500 ms post-stimulus onset represents mainly auditory activity and potential maintenance and memory loops. This exclusion should be better justified and either added to or discussed in the main text. Adding this ROI could help to tease apart primary visual and auditory activity and their interactions in space, as a complement to the provided temporal results.

In general, the motivation for focusing on these four very specific ROIs is rather short and could be expanded (lines 703 – 708). Mentioning the reasoning clearly up front in the introduction would help to guide the reader too.

Interactions between primary areas are being mentioned as a key question of the investigation throughout the manuscript. However, most neural analyses examined each ROI separately. Interactions were not quantified by means of granger causality, mutual information or some other

method. Without such added analyses the authors' conclusion rests merely on the time points of the decoder performance and a descriptive change in sensitivity and criterion.

Line 223 and methods: Starting with figure 3C and D it would help the reader to specify clearly which 1.5 seconds of the procedure these results are referring to. This should be done in the figure captions and decoder section of the methods.

Line 264: Training the classifier in visual and audiovisual trials might average out idiosyncratic features indeed, however, how the decoder behaves in primary visual and auditory cortices separately could shed light on potential early (< 400 ms) interactions between these regions, as it should be more specific in these areas.

To address the questions of whether enhancements in sensitivity act on early, late or both perceptual stages, the manuscript would benefit from showing scalp topographies as decision components have frequently been shown to have characteristic activity patterns (e.g., a strong centro-parietal focus of the "late" decision component). Scalp topographies would also help to clarify what neural activity was present around the important peak time point of 500 ms. Scalp topographies could also shed more light on whether "post-sensory visual information" is actually maintained (e.g. line 531), as this is an often used term for evidence accumulation processes in perceptual decision-making.

Line 202: Calling a window spanning 400 – 600 ms an early window might be confusing given established/traditional perceptual decision-making terminology where time points up to 300/400 ms post-stimulus onset are often considered early. Similarly, terming the 900 to 1100 ms time window late is rather uncommon, as many other decision-making studies term decision components around 500-700 ms post stimulus onset late. This window often represents the end of the evidence accumulation process. Particularly, when stimuli are only presented very briefly. Elsewhere in the manuscript (line 272) the authors speak of "late perceptual information (500 ms)", which is not in line with the earlier description. I suggest aligning the manuscript's temporal terminology with the terminology most common in the field and being consistent throughout the manuscript. This would help its clarity.

The authors do not attempt to make much use of the temporal information of their results by merely describing time points but without linking them to functional relevant processing stages. Therefore, I would like the authors to elaborate more on the functional meaning of their temporal decoding results. This is important, since one of the main research questions revolves around differentiating between perceptual and decision level (biases) processes that can also be dissociated in time (early vs late) and space. This could further strengthen the neuro – behaviour link.

Moreover, the authors favour the explanation of a late maintenance model in which a decision variable is storing relevant information until the response. A response-locked analysis locked to the onset of the decision phase might help to corroborate this explanation, if the traditional parietal ramp up activity of the decision variable is maintained or shows a plateau in amplitude until the subsequent decision phase.

A more detailed description of the statistical analysis, particularly the key cluster-based permutation analysis, would be helpful for the reader and necessary to be able to reproduce it. The presented paragraph at the very end of the methods is rather short given such a complex and important analysis. It leaves the reader wondering how reported p values were obtained. Usually cluster t-values of the permutation distribution are used to define data-driven significance thresholds as in the original publication by Maris and Oostenveld (2007). These clusters would be obtained in space and time to correct for multiple comparisons. Here, it seems that the authors only corrected for multiple comparisons in the spatial domain but not the temporal domain (i.e., number of consecutive significant temporal samples). This may not be a flaw of the analysis but more of the description. However, if the temporal domain wasn't considered this should also be included in the cluster analysis. I'd invite the authors to clarify this part of their analysis.

Figure captions of figures 3-8 state that significant clusters are depicted by thin red markings.

While these are very hard to see they also appear patchy and small in their extent. Following my previous point, this might suggest that they would not survive correction for multiple comparisons in the spatial and temporal domain. The temporal windows described in the text are not presented in the decoding figures, their addition (as in S11) might help the reader. Please disregard this comment if this is merely an issue with visibility of the markings and consider making them more visible.

Some result statements about the absence of an effect in these data (i.e., tests that yielded a p value above the arbitrary frequentist significance threshold of $p = .05$) cannot be made with such strengths using the frequentist framework (e.g., line 143-144 and 147-149). The reason being that absence of evidence is not evidence of absence. To support these statements, I'd suggest that the authors use Bayesian statistics and Bayes Factors to be able to make statements on the likelihood of their data under the null hypothesis (absence of an effect).

The discussion would benefit from discussing the link between the temporal results and of the decoder and their functional interpretation in more detail. It would also benefit from discussing what the effects of other auditory stimuli might have been, as bottom-up sound-driven modulations of early visual cortex are a key finding. That is, would more complex sounds trigger activity in other parts of the visual cortex further up the hierarchy (e.g. extrastriate cortex) and not in early visual cortex?

Minor concerns

What was the motivation for choosing a set of very bright/salient colours for experiment 2? For example, magenta might be more salient than yellow. I presume it's unlikely that the order of the colours might have biased the results substantially, but I would ask the authors to clarify how the colour sequence was picked and randomized for each trial.

The authors should consider showing the behavioural results of experiment 2 in the main text and not only as a supplementary figure (currently S3), since the absence of behavioural effects given the contrast level and modality is an important result for their claims. Further, results depicted in figure S5 and S6 are being mentioned in line 179 as prior evidence. I think these are deserving of being incorporated in a figure of the main manuscript.

The supplement shows two figures labelled S5.

Please use the same scale for all d_{prime} and all criterion temporal generalization plots in figs 4-7. Currently the colourmaps are varying in range.

Line 147: please specify more clearly what a "more liberal response threshold" actually means in the context of your task.

Line 347: It would be informative to report accuracy values alongside d_{prime} .

Line 612: How many participants or blocks required an adjustment due to ceiling or floor effects? Please add this information, if available.

I outline several suggestions for phrasing and typos:

- Line 64: Most previous neuroimaging (delete of)
- Line 115: This is in contrast
- The sentence in line 207 is truncated, please double check the ending
- Line 251: information conformed to a continuous pattern
- Line 268: the word between is redundant
- Line 398: false alarm reports
- Line 457: their feature-based attention
- Line 502: the percept of an additional
- I believe the duration in line 629 should read 33ms instead of 32ms.
- 706: involved in short-term memory

We thank both reviewers for their insightful comments. We have adapted the manuscript accordingly, and believe the present submission is a substantial improvement over the previous. Specifically, we would like to highlight that following the reviewers' suggestions 1) we have clarified those sections that required further explanation, 2) we have performed additional control analyses to demonstrate the correct regularization of the covariance matrix used in the source-reconstruction analyses, 3) we have made more explicit the main goals of this research, justifying the inclusion of the four selected regions of interest 4) and we have performed a new analysis that have helped to better understand how sound enhanced visual detection. These new results are now included in the final version of the manuscript.

Below, we go into each of the comments in detail.

Reviewers' comments:

Reviewer #1 (Remarks to the Author):

This is a fascinating study on crossmodal interaction (audiovisual) using temporal generalization analysis of source-space MEG data. The authors first measure the sensitivity (d') and criterion I using behavioral analysis and then use Latent Discriminant Analysis (LDA) to derive the same from MEG data. The sensitivity indicates the ability to discriminate between two conditions while the criterion score reflects the threshold for detection. Lower criterion leads to more false alarms. The authors show that these parameters derived from behavior and experiment are correlated.

Next, two experiments are conducted to answer specific neuroscientific hypothesis. In the first experiment, the authors test the hypothesis that sounds enhance detection due to crossmodal interactions and increase the false alarm rate (lower criterion). In the second experiment, the authors test the hypothesis that crossmodal interactions are enhanced due to attention and whether or not decision-level bias plays a role in the enhancement. While I cannot comment on the audiovisual integration literature, I find the machine learning analysis that has been used to probe psychological measures innovative. Specific comments are noted below:

Major points

1. Given that the correlation between behavioral and decoded criterion parameters is extremely low (around 0.1; Fig 3D), how reliable are the conclusions in the other figures? It is also not clear how many values are these correlations computed over? Are they for 24 subjects? Ideally a scatter plot of behavioral vs decoded psychological scores with 24 points will clearly show how reliable the proxy measures are.

Following the reviewer's recommendation, we have graphically represented how the correlations between behavior and classifier SDT parameters are computed. In Fig. R1B/ Fig. S4B we show the d' prime and criterion correlation analysis for one representative participant at two different time points (0 s and 0.5 s). In short, we correlated individual behavioral d' primes with the d' primes estimated by the classifier across the 6 experimental blocks. Thus, given that we tested 24 participants, for each training/decoding time combination there are 24 correlation values. The maps

in Fig 3C show the correlations averaged across participants. In Fig. R1C/S4C, we show the group level correlation distribution at 0 and 0.5 s. Whereas participants d 's are heavily correlated with the d 's predicted by the classifier at 0.5 s, the d 's correlation is not significantly different from 0 at 0 s. Correlations between participants criterion and the classifier predictions are weaker compared to the d 's correlations, but still are significant at 0.5 s but not at 0 s. We have described with more detail the analysis pipeline at the methods section and added a representative figure to the supplementary materials (included below as well for the reviewer's convenience).

In relation to the low correlation values for the criterion condition, we have realized that there was a mistake in the "colorbar" scale values. Instead of going from 0.1 to -0.1 it goes from 0.3 to -0.3. We apologize, and have corrected this in the manuscript.

Figure R1. A Temporal generalization matrixes with d' and c correlation values in the first experiment. Blue and red crosses indicate the two time points of interest (0 to 0.5 s) used in the example correlation analysis. B Example of d' and c correlation analysis (upper and bottom panels respectively) for one representative subject at two different time points. Whereas the behavioral and decoded d' and c parameter are positively correlated at 0.5 s, the correlation between behavioral and neurally decoded data at time 0 s is not different from 0. C Group level distribution of correlation estimates at time 0 (x axis) and time 0.5 (y axis) for d' and c parameters. Each color point represents one participant.

“Correlation between neurally decoded and behaviourally estimated SDT parameters

We used the LDA classifier predictions to generate the TGMs with the d' and c parameters for each one of the six experimental blocks. To test whether the neurally decoded parameters were associated with the participants performance, we correlated the six neurally decoded d 's with the six behaviorally estimated d 's in each TGM train-test temporal combination (Fig. S4). The same correlation procedure was repeated on the criterion parameters. We ended having 24 correlation maps (one per participant) for each SDT parameter. Using a cluster-based permutation approach

we tested in which decoding train-test combination of the TGM the correlation distribution between the neurally and behaviorally estimated parameters was different from 0.”

2. The time generalization matrices look suspiciously similar between different brain regions. Have the authors verified that this is not an artifact of the data processing pipeline? It might be helpful to show in a figure: a) the source localization on the cortex at the time instant that the evoked response peaks, and b) the virtual channels for one condition and subject in the four ROIs being analyzed. It will verify that the noise covariance was correctly regularized and noise was effectively suppressed when computing the source estimate.

We thank the reviewer for this suggestion and would like to start by emphasizing that we expected to find a reasonable decoding overlap across the different ROIs, given that: 1) All the selected ROIs are involved in broadcasting and maintaining visual information 2) Although source-reconstruction spatially suppresses the activity of distant sources, there might be some residual activity leaking between different ROIs virtual channels. A classifier can pick up on these weak but hypothetically consistent activity fluctuations and decode the same information in multiple brain regions. Nevertheless, although the decoding patterns are approximately similar across ROIs, the effect of sounds on information decoding point towards meaningful functional differences in visual detection between the selected ROIs. For instance, whereas sounds enhanced the late maintenance of visual information in the visual and inferotemporal ROIs, the encoded information barely generalizes in the parietal ROI, and interestingly, in the DLPFC ROI the sound-induced decoding enhancement takes place just before the beginning of the response phase, pointing towards a functional involvement in the late reinstatement of information in visual cortex. We believe that these ROI based analyses are informative to define future research hypotheses on this topic.

Following the reviewer’s recommendation, we source-reconstructed activity in the visual and in the auditory conditions to validate the correct regularization of the covariance matrix.

Figure R2. A. Visual (0.2 to 0.3) and B. auditory (0.1 to 0.2 s) source-reconstructed activity projected in a surface model of the brain. Same results are plotted in a standard MNI template for the visual C and the auditory D activities.

In Fig. R2A and R2C we contrasted source-reconstructed activity in S+ vs S- trials at 0.2 to 0.3 s (taking visual and audiovisual trials together). As expected, the strongest (early) activity modulation is measured at occipito-parietal regions. In figure R2 B and R2 D we also source-reconstructed (early) auditory activity by contrasting auditory (A+) and visual (A-) trials. The largest activity modulations are found bilaterally in temporal regions overlapping with Heschl gyrus (i.e. superior temporal cortex).

Figure R3 Individual (A) and group average (B and C) of the event-related field potentials (ERFs) projected through the virtual channels conforming each ROI (presented in different columns). The continuous lines represent activity in the S+ trials and the dashed lines represent activity in the S- trials. Panel A represents source-reconstructed activity for one subject that participated in the first experiment. In panel B and C, group average activity of all the participants in the first experiment and second experiments respectively (in experiment 2, S- and S++ conditions).

In Fig. R3 we show the ERF activity projected over the virtual channels conforming each ROI. In Fig. R3A, we show the ERF source-reconstructed activity for one representative participant. The MEG activity profile of individual participants is highly similar to the group average MEG activity profiles. Thus, here we will interpret the less noisy signal conveyed by the group level averages represented in Fig. R3B and R3C.

As expected, the largest visual modulations are measured at the visual cortex (e.g. place-holder disappearing at 1.25 s). The visual ROI is also highly modulated by sounds, in agreement with previous studies (Brang et al., 2015; Feng et al., 2014). This sound-induced modulation of visual cortex activity is larger when participants are explicitly engaged in a visual detection task (Experiment 1 in Fig. R3B) and the sounds are informative for the task. The remaining ROIs show very weak modulations induced by the presentation of visual or auditory stimuli.

In summary, these results show the expected response profile in each brain regions and ROI to the different types of stimuli and demonstrate that the covariance matrix is correctly regularized. We have updated Fig. S1 to demonstrate that visual and auditory activity is correctly source-reconstructed (see Fig. S1B and below):

Figure S1. A Grid points conforming each ROI overlaid on two different views of a MNI standard template headmodel (Visual cortex ROI: green points, Inferotemporal cortex: yellow points, Parietal cortex: blue points, Dorsolateral prefrontal cortex: red points). B. We validated that source-reconstructed activity was correctly localized by overlaying group level visual (0.2 to 0.3) and auditory (0.1 to 0.2 s) source-level activation on a surface model of the MNI standard template. The activations were obtained by subtracting S- from S+ trials (visual activity) and A- from A+ trials (auditory activity) in Experiment 1. As expected, early visual activations were localized in visual regions (occipital and parietal cortices) and early auditory activations in auditory regions (superior-temporal cortex). C. Decoding sensitivity in Experiment 1 (visual and audiovisual modalities together) for each ROI at matching training and decoding time points (diagonal of the TGM): Information decoding peaks at 500 ms and nicely follows the temporal ordering expected for each ROI as a function of its level within the perceptual hierarchy.

3. Related to the above point, have the authors considered what the upper triangle (where differences are observed in Figure 4 for example) of the TGM means? It seems to indicate that neural processes later in time can explain neural processes earlier in time pointing to an acausal phenomena.

We apologize for not being clear here. TG analyses do not aim to infer causal directionality. What the mentioned pattern of results means is that a classifier trained to decode S+ from S- trials at the end of the trial can still decode information at the beginning of the trial because the same information still remains in both time points. The asymmetry in the TGM is understood by noting that the train and test trials are not fully exchangeable. Specifically, for this analysis, we trained on all AV+V trials, while we tested separately on the AV and V trials. We conducted a control analysis to explore this asymmetry, and write about this on page 13 in the manuscript:

“In a complementary control analysis (Fig. 4C and Fig. S8), we confirmed that by training and testing the classifier in the visual and audiovisual conditions separately, the sound-induced d' enhancement pattern became symmetrical with respect to the diagonal.”

4. Is there a bias in the LDA classifier somewhere because the baseline period in Figure S4 does not show 0.5 as the hit rate?

This behavior from the classifier is expected considering that the classifier has been trained using data at 0.5 s, at which point there is S+ information. Therefore, in those points in which the signal has not been yet presented (e.g. baseline) the classifier will be less likely to report S+ (hit or false alarm), because no signal information is present. It is important to note that the classifier is not biased to report more signal in the S+ compared to the S- conditions ($pHits = pFAs$), as evident by a mean criterion = 0.

Minor points

1. The inverse of the covariance matrix has been called the covariance matrix in the methods section

Thanks. We have corrected the paragraph “Using the (inverse) covariance matrix computed from the combined visual and audiovisual trials (-0.2 to 1.5 s, time-locked to the target onset; 10% regularization), the volume conduction model, and the lead field, we applied a linearly constrained minimum variance (van Veen et al., 1997) beamformer approach in Fieldtrip (Oostenveld et al., 2011) to build a common spatial filter for each grid point and participant.”

2. The use of tilde is not clarified. Notations need more clarity and consistency. What is tilde?

I used tilde to indicate that the described quantity was approximate (not exact). I have removed or substituted the tilde notation along the manuscript.

3. The quality of certain figures is low. They appear to be screenshots rather than high resolution images for pdf

We apologize for the low resolution in the initial submission. As noticed by the reviewer, the figures attached in the first version of the manuscript were simple screenshots. We have now submitted the manuscript including high quality vectorized images of the same figures.

4. Signal Detection Theory (SDT) abbreviation needs expansion before use

Done

5. Authors refer to “folds” as “manifolds”. Manifolds has a specific meaning in mathematics which is perhaps not what they mean in the context.

Thanks, we have corrected the term along the manuscript.

6. References for ICA and spatial interpolation are missing

Reference added for ICA: (Vigário et al., 1998). The spatial interpolation is a simple, standard, procedure, implemented in FieldTrip, which we now describe and reference as follows: “Bad channels showing SQUID jumps or other artifacts were interpolated to the weighted mean of neighboring channels. Weights correspond to

the physical distance between sensors (ft_repairchannel, as implemented in FieldTrip, Oostenveld et al., 2011)."

7. Expand abbreviation SM to supplementary material

Done.

8. "surface of the volume conduction model" seems like a weird phrase

Rephrasing: "external boundary of the volume conduction model"

9. References for visual sensitivity and criterion parameter are not fully appropriate as none of the references actually explain what these parameters are or how they are computed. Alternatively, the authors should explain this in the methods section and refer to the methods section.

We agree with the reviewer that the potential readers of Communications Biology might not be familiar with Signal Detection Theory parameters interpretation. We have clarified the computation and interpretation of d' and c parameters at the methods section and referred to it in the main text.

"Observers' sensitivity (d' ; i.e., the ability to distinguish signal from noise; see methods) in detecting the visual target was higher in the audiovisual compared to the visual conditions...."

"In the first experiment, we quantified participants sensitivity (d' ; i.e., the ability to distinguish signal from noise) to discriminate S+ from S- trials using the SDT (yes-no paradigm). In addition, to estimate participants biases to report S+ or S- independently from their visual sensitivity, we computed the criterion (c ; i.e., (negative) bias) parameter (Macmillan & Creelman, 2004). SDT sensitivity and criterion parameters were calculated for each condition and block by coding S+ trials correctly reported as "signal" as hits and S- trials incorrectly reported as "signal" as false alarms. This way, lower c parameter values represented a more liberal criterion to report the signal as present. In the second experiment the participants task was to remember a sequence of items and report whether the third item was the same or different from the previously presented items. Therefore, we again applied SDT to calculate participants sensitivity, but assuming a same-different "independent observation" model paradigm. Here, high d' values represented better performance in discriminating whether the last fixation point color had been presented or not during the trial sequence. Complementarily, the c parameter indexed the participant's bias to report the last fixation point color as repeated (same) or different."

10. In several parts of the text, the authors refer to the supplementary material. In my opinion, the main text should be readable without supplementary figures and references to them should be sparingly used. It breaks the flow of the reading.

We thank the reviewer for this suggestion and agree. We have reduced the number of references to supplementary materials by integrating some supplementary figures to the main text (Fig. 3C and Fig. 7). Now, the figures and tables included in the

supplementary materials correspond to control analyses and/or specific details about the methods that are not necessary to understand the main results of the study.

Reviewer #2 (Remarks to the Author):

Thanks for giving me the opportunity to review your manuscript entitled “Seeing sounds: Neural mechanisms underlying auditory contributions to visual detection”. In this manuscript, Perez-Bellido et al. use two complementary audiovisual paradigms to show that an additional sound played alongside a visual grating can facilitate detection and the presence of the grating can be decoded from neural activity using a decoder trained on visual and audiovisual trials together. These enhancements in sensitivity come with a concomitant criterion shift whose origin appears to be dependent on top-down controlled attention, as this shift wasn’t present in experiment two. The authors interpret their results as broad effects of the integration of audiovisual stimuli across the entire processing hierarchy in time and space (ROIs).

Overall, this investigation of the mechanisms underlying the neural integration of audiovisual signals, their point in time, and effects of top-down control is timely. Multiple groups have attempted to shed light on these questions in recent years, but important questions remain to be answered. The present manuscript benefits from combining a traditional visual detection task with simple auditory tones. This is different from some previous paradigms, which used only simple or rather complex stimuli in both modalities. The focus on signal detection theory for quantifying the characteristics of the neural decoder is elegant and less common in the field. The pirate plots of behavioural and signal detection results depict these data and their different elements well. Taken together, while I appreciate the nice design of the two complementary experiments, the incorporation of visual eccentricity, and the manuscript’s well-written style, I think the present version would benefit from further evidence to deliver on its ambitious goals. To foreshadow my recommendations, the temporal and scalp map results should be explored further by linking them to a functional interpretation more strongly and the statistical analysis would benefit from some considerations to corroborate the authors’ claims unequivocally.

We appreciate the reviewer positive assessment of this manuscript. Below we will address the raised concerns to the best of our ability.

Specific comments with recommendations:

Major concerns

1a. Given that one part of the authors question was to investigate where potential enhancements of auditory sounds would take place in the brain (e.g., early effects due to connections between primary visual and auditory cortices), I was surprised to see that the ROIs did not include an additional ROI for primary auditory areas (superior temporal lobe). The results presented in figure S12 suggest to me that the information encoded at the peak time point 500 ms post-stimulus onset represents

mainly auditory activity and potential maintenance and memory loops. This exclusion should be better justified and either added to or discussed in the main text. Adding this ROI could help to tease apart primary visual and auditory activity and their interactions in space, as a complement to the provided temporal results. In general, the motivation for focusing on these four very specific ROIs is rather short and could be expanded (lines 703 – 708). Mentioning the reasoning clearly up front in the introduction would help to guide the reader too.

The reviewer rightly highlights the importance of understanding the neural mechanisms supporting early audiovisual interactions. However, we would like to emphasize that in this specific study, we were mainly interested on understanding how sounds change the processing of visual information in a visual detection task. Specifically, we wanted to characterize the neural mechanisms supporting auditory driven changes in visual sensitivity and criterion. Thus, our interest was to specifically focus on those brain regions engaged in visual detection processes. For that reason, we just included four ROIs that have been typically associated with visual processing and detection awareness. In line with the reviewer's suggestion, we are now more explicit about the objectives of this project in the introduction. We have also included a more explicit description of why we have selected those four specific ROIs already at the beginning of the manuscript.

“Because the goal of this study was to characterize how sounds modulate the neural processes underlying changes in visual sensitivity and decision criterion, we constrained our decoding analyses to four source-reconstructed brain regions of interest (ROIs) engaged in the broadcasting and maintenance of stimulus information in visual detection tasks 42,43. These were the visual cortex, that is involved in the encoding of sensory information (Ress & Heeger, 2003), the parietal cortex, that is related to evidence accumulation during perceptual decisions (Kiani & Shadlen, 2009; Zhou & Freedman, 2019), the inferotemporal cortex, that has been associated to visual memory and target identification (Miller et al., 1993; Mishkin, 1982; Pagan et al., 2013) and the dorsolateral prefrontal cortex, that is involved with short-memory, decision-making and awareness (Funahashi, 2006; Kim & Shadlen, 1999; Philiastides et al., 2011; van Vugt et al., 2018).”

The reviewer suggests that the information encoded at the peak time point 500 ms post-stimulus onset represents mainly auditory activity and its potential maintenance in memory loops. Although it is true that there is some residual auditory activity at 500 ms, we believe that it is unlikely that the information decoded at time point 500 ms is auditory, given that the classifier was not trained to decode auditory activity. In addition, S+ and S- folds used to train the classifier to discriminate between S+ and S- trials were balanced in number of audiovisual trials, making both visual (present/absent) and auditory (present/absent) conditions orthogonal to the classification.

2. Interactions between primary areas are being mentioned as a key question of the investigation throughout the manuscript. However, most neural analyses examined each ROI separately. Interactions were not quantified by means of granger causality, mutual information or some other method. Without such added analyses the authors'

conclusion rests merely on the time points of the decoder performance and a descriptive change in sensitivity and criterion.

We agree with the reviewer that broadening our understanding of the mechanisms by which early sensory areas exchange information is a relevant question. Indeed, as discussed in the main text of the manuscript, there is ample evidence for the existence of cortico-cortical direct connections between auditory and visual regions, and these might be at the base of many early crossmodal interactions. In the interest of maintaining a consistent scientific report, we have focused on characterizing how sounds modulate sensitivity and criterion in visual detection, emphasizing the visual and visuo-cortical consequences of possible audiovisual effects. This already encompassed a rather large set of behavioural and neural analyses, based on two distinct experiments. While exploring the functional connectivity between auditory and visual cortex would be interesting, this lies beyond the scope of the present study (which was also not designed with the analysis of functional connectivity in mind).

We do appreciate that, because we do not directly present any results regarding functional connectivity, some of the objectives of the study and conclusions regarding such connectivity may have to be toned down. Therefore, we have adapted:

Experiment 1

“This manipulation was motivated by previous neuroanatomical tracing studies in monkeys showing that more peripheral visual eccentricities receive denser projections from primary auditory cortex 53,54, and it allowed us to verify that the mechanisms involved in audiovisual integration did not depend on visual eccentricity 12,55.”

Discussion

“Retrograde tracing studies in monkeys have revealed the existence of direct connections between the auditory the visual cortex. These connections are heterogeneously distributed across the visual field with more peripheral eccentricities receiving denser projections 54,74. Given that such asymmetrical connectivity pattern could determine the mechanism by which sounds interact with visual detection, in our first experiment we tested the effect of sounds on visual stimuli presented at central and peripheral visual eccentricities. Our behavioral and neural decoding analyses in the first experiment yielded converging results at both eccentricity conditions, allowing us to conclude that the mechanism by which sounds enhanced visual detection in this study does not depend on retinal eccentricity. Future studies should further investigate whether the heterogenous auditory-to-visual cortex connectivity pattern reported in monkeys replicates in humans 83, and if it actually involves a functional dissociation for multisensory processing.”

3. Line 223 and methods: Starting with figure 3C and D it would help the reader to specify clearly which 1.5 seconds of the procedure these results are referring to. This should be done in the figure captions and decoder section of the methods.

We thank the reviewer for this suggestion. We have now explicitly declared the temporal window of interest in the methods and decoding sections. We have also characterized what the TGM vertical lines represent in the figure caption (stimulus onset and response phase latencies).

4. Line 264: Training the classifier in visual and audiovisual trials might average out idiosyncratic features indeed, however, how the decoder behaves in primary visual and auditory cortices separately could shed light on potential early (< 400 ms) interactions between these regions, as it should be more specific in these areas.

Following the reviewer's advice, we performed an additional control analysis. We source-reconstructed ERF activity using an anatomically defined auditory ROI, and using an information decoding approach, we tested at which temporal latencies sounds modulate brain activity patterns. We trained a classifier to decode auditory information separately in the S+ and S- conditions. We followed the same methodological TG approach used to decode visual information, however here we decoded auditory present from auditory absent (A+ from A-) trials.

Figure R4 Temporal generalization matrixes represent the decoder's sensitivity in decoding auditory stimuli from an auditory ROI in experiment 1 (A) and experiment 2 (B) for the S+ and S- (left and right panels respectively). C and D represents the difference in auditory decoding between the S+ and S- conditions in the first and second experiments respectively.

Suprathreshold auditory stimuli evoked strong MEG signal modulations in auditory cortices. This can be readily picked up by the classifier at much earlier latencies than the visual signals (around 30 ms after the stimulus onset). In fact, the auditory decoding peak also appeared earlier in time (around 100 ms) compared to the visual target decoding peak (around 500 ms), and the sound-driven activity pattern perturbation was very stable in time (i.e. strong generalization of decoding over time). It is important to note that although the auditory stimuli were rendered task irrelevant in both experiments, sounds induced larger activity modulations in experiment 1 compared to experiment 2 (Fig. R.3A and B). This difference in activation strength is also captured by the classifier with higher auditory decoding sensitivity in experiment 1 compared to experiment 2 (Fig. R4).

These results reinforce the hypothesis that in the first experiment, participants actively relied on the auditory stimulus to guide their temporal attention in a top-down fashion. Accordingly, in Fig. R3C we observed that when participants detect the visual signal (S+ trials) at around 500 ms, the classifier auditory decoding sensitivity is significantly reduced compared to S- trials. This suggests that once the participants are processing the visual stimulus, their attention towards the auditory stimuli (and thus the associated neural signal) was reduced.

Conversely, in the second experiment the participants did not have to detect the visual stimuli and therefore, the grating presentation did not reduce the attention towards the auditory stimuli. This is represented by the null difference in auditory decoding sensitivity between S+ and S- conditions in the second experiment Fig. R3D.

We consider that these control analyses provide indirect evidence for our original conclusion postulating “that participants strategically used auditory information to guide their attention in time”. Given that this new analysis does not change our current interpretation of the results, and only provides indirect supporting evidence for this hypothesis, to reduce the complexity of the manuscript we have decided to leave these results out of the manuscript.

5. To address the questions of whether enhancements in sensitivity act on early, late or both perceptual stages, the manuscript would benefit from showing scalp topographies as decision components have frequently been shown to have characteristic activity patterns (e.g., a strong centro-parietal focus of the “late” decision component). Scalp topographies would also help to clarify what neural activity was present around the important peak time point of 500 ms. Scalp topographies could also shed more light on whether “post-sensory visual information” is actually maintained (e.g. line 531), as this is an often used term for evidence accumulation processes in perceptual decision-making.

We thank the reviewer for this helpful suggestion. To test the hypothesis proposed by the reviewer (e.g., a strong centroparietal focus of the “late” decision component), we calculated the average (planar combined) ERF activity evoked by the visual target in the V and AV conditions. First, we plotted the S+ minus S- scalp topographies at 500 ms, coinciding with the target decoding peak (Fig. 5A). Interestingly, whereas the topography in the visual condition showed a spatially distributed and heterogenous pattern of activity, the topography in the audiovisual conditions showed that the activity modulation was constrained to centroparietal and occipital sensors.

Second, to explicitly test the prediction formulated by the reviewer, we contrasted the temporal evolution of centroparietal activity in the visual and the audiovisual conditions. To do so, we averaged the ERF activity over several centroparietal MEG sensors (Fig. 5B). The activity registered in these sensors indexes the centroparietal positivity (CPP), a relevant neural signature of evidence accumulation in perceptual decision-making processes (O’Connell et al., 2012; Twomey et al., 2016). This analysis showed that centroparietal activity unfolds very differently in the visual and audiovisual conditions: Whereas in visual trials centroparietal ERF activity ramps

steadily and remains active until during most of the trial length, in the audiovisual trials, the activity shows an early reset on evidence accumulation (around 250 ms), and suddenly ramps up very strongly (Fig. 5C), surpassing the visual CPP at around 500 ms. These results suggest, as we already proposed in the previous version of the manuscript, that sounds enhance visual detection probably by optimizing the process of evidence accumulation. This enhancement might be mediated by focusing more “attentional” resources on processing those perceptual samples in which the target stimulus is more likely encoded. This hypothesis is also supported by the scalp topographies showing a more specific and probably efficient engagement of visual and centroparietal regions during the decision process in the audiovisual compared to the visual conditions.

We believe that these new results help to understand how sounds modulate perceptual decision-making. Therefore, we have included the new figure 5 and related results and methods sections in the manuscript.

We have also updated the main text describing:

Introduction: “Finally, to explore how sounds changed visual evidence accumulation during visual detection we supplement our decoding analyses with a sensor-level analysis of parietal activity.”

Figure 5 A Difference between the sensor-level ERF activity in the S+ and S- conditions measured at (A) whole-brain scalp topography, and (B) centroparietal sensors (red points in the topography inset). C Evidence accumulation gain, measured as the slope of the differential ERF activity in B, as a function of time using a 200 ms sliding window. Black points in A depict those MEG sensors with significantly different activity in the S+ and S- conditions. Line contours depict standard error. Red and green rectangles compress temporal clusters in which there are significant differences between the S+ and S- conditions. Black rectangles depict temporal clusters in which there are significant differences between the visual and the audiovisual timeseries. Statistics are obtained using cluster-based permutation tests.

Results:

“Sounds enhance the gain of post-sensory evidence accumulation

The d' parameter decoding analyses showed that sounds enhanced the maintenance of late perceptual representations. This enhancement might be preceded by a sound-induced improvement in evidence accumulation (Franzen et al., 2020). To test this hypothesis, we contrasted the unfolding of centroparietal event-related field (ERF) activity in the visual and the audiovisual conditions. Previous studies have shown that electrophysiological activity in centroparietal sensors correlate with changes in evidence accumulation in perceptual decision-making tasks (O'Connell et al., 2012; Ratcliff et al., 2009). Thus, if sounds enhanced the accumulation of perceptual evidence, we expected to find a larger positive modulation of centroparietal activity in the audiovisual compared to the visual trials. We first characterized the scalp topography of the (planar-combined) ERF activity evoked by the visual targets at the decoding peak (500 ms) in the visual and audiovisual conditions. We found that whereas in the visual condition, the processing of the visual target induced a broadly spatially distributed activity pattern across the scalp topography (Fig. 5A), the processing of the visual target in the audiovisual condition was spatially constrained to visual and centroparietal sensors. To better understand how sounds modulated centroparietal activity, we limited our analyses to a subset of MEG sensors localized in centroparietal regions (see methods). This analysis showed that centroparietal activity unfolds very differently in the visual and audiovisual conditions: Whereas centroparietal activity increased slow but steadily in the visual condition, on the audiovisual condition the activity was initially reduced (at around 350 ms), and then it quickly ramped up (Fig. 5B). To quantify the differences in evidence accumulation rate between both conditions, we calculated the slope of the centroparietal activity modulation as a function of time using a 200ms sliding window (Fig. 5C). In consistence with the previous analysis, we observed that the activity gain from 250 to 350 ms was significantly reduced in the audiovisual compared to the visual trials, however, from 350 to 500 ms the activity gain was sharply reversed experiencing a strong increment compared to the visual condition. These results demonstrate that evidence accumulation in the visual condition proceeds more gradually than in the audiovisual condition. However, the integration of evidence in the audiovisual conditions evolves later but much faster. It is possible that sounds helped the participants to condense their processing resources on those temporal latencies in which the target information is more readily available, boosting the efficiency of the evidence accumulation process and subsequently leading to a better encoding and maintenance of decision information. This hypothesis is supported by the scalp topography in the audiovisual condition that in contrast with the visual condition, revealed a more efficient recruitment of those brain regions specialized in perceptual decision-making (i.e. the visual and parietal cortices) at the same time that contained more decodable target information."

Discussion

"Based on the temporal ordering of the significant clusters across ROIs (Fig. S12) and in line with the currently accepted view that multisensory interactions depend upon a widely distributed network of brain regions 4,64–66, we propose that a late maintenance dynamical mechanism could be implemented, first by the encoding of the visual stimulus in the visual cortex. Concurrently, the encoded sensory information would be accumulated into a latent DV in parietal regions 43,44,52. This

is a stochastic process in which, as demonstrated by the centroparietal activity analyses, an auditory cue can optimize by refocusing the participant's processing resources towards those sensory samples that more likely encode the visual target."

And in other parts of the Discussion section.

6. Line 202: Calling a window spanning 400 – 600 ms an early window might be confusing given established/traditional perceptual decision-making terminology where time points up to 300/400 ms post-stimulus onset are often considered early. Similarly, terming the 900 to 1100 ms time window late is rather uncommon, as many other decision-making studies term decision components around 500-700 ms post stimulus onset late. This window often represents the end of the evidence accumulation process. Particularly, when stimuli are only presented very briefly. Elsewhere in the manuscript (line 272) the authors speak of "late perceptual information (500 ms)", which is not in line with the earlier description. I suggest aligning the manuscript's temporal terminology with the terminology most common in the field and being consistent throughout the manuscript. This would help its clarity.

We apologize for any confusion. To alleviate this, we have modified the adjectives. Instead of using "early" and "late" clusters now we refer to them as "first" and "second" clusters. We have aligned the temporal terminology along the manuscript by considering effects taking place < 250 ms as early effects, and effects taking place > 250 ms as late effects, in consistency with previous studies.

7. The authors do not attempt to make much use of the temporal information of their results by merely describing time points but without linking them to functional relevant processing stages. Therefore, I would like the authors to elaborate more on the functional meaning of their temporal decoding results. This is important, since one of the main research questions revolves around differentiating between perceptual and decision level (biases) processes that can also be dissociated in time (early vs late) and space. This could further strengthen the neuro – behaviour link.

We thank the reviewer for this suggestion. We have now added more temporal references to the functional interpretation of the role played by the different ROIs in the perceptual decision-making process:

"... we propose that a late maintenance dynamical mechanism could be articulated, first by the encoding of the visual stimulus in the visual cortex. Concurrently, the encoded sensory information would be accumulated into a latent DV in parietal regions (Kiani & Shadlen, 2009; O'Connell et al., 2012; Zhou & Freedman, 2019). This is a stochastic process in which, as demonstrated by the centroparietal activity analyses, an auditory cue might amplify the participant's gain towards those DV samples that more likely encode the visual target. The enhance DV accumulation is maximal around 500 ms, and the relevant decision information is stored in the inferotemporal and dorsolateral prefrontal cortices to protect them from interference with new incoming sensory input. Indeed, the interplay between these two brain regions has been demonstrated to play an important role in short-term memory during perceptual decisions (D'Esposito & Postle, 2015; Gross et al., 1972; Pagan et al., 2013; Stokes et al., 2013; van Vugt et al., 2018). As the response phase approaches, the participants can strategically reorient their feature-based attention

towards the stored relevant stimulus information. This manifests by an enhanced reactivation of the target at 1000 ms in the DLPF cortex (Squire et al., 2013), that subsequently leads to an enhanced reinstatement of the stimulus information in early visual regions (Christophel et al., 2017; D'Esposito & Postle, 2015; Sprague et al., 2016)."

8. Moreover, the authors favour the explanation of a late maintenance model in which a decision variable is storing relevant information until the response. A response-locked analysis locked to the onset of the decision phase might help to corroborate this explanation, if the traditional parietal ramp up activity of the decision variable is maintained or shows a plateau in amplitude until the subsequent decision phase.

In our experimental design, the response phase always starts 1.25 s after the stimulus onset (vertical and horizontal dotted lines in the TGMs). Therefore, it is important to note that any stimulus-locked analysis is also response-phase-locked; there was no stimulus of extended duration about which evidence could be accumulated until a response. Keeping this in mind, by looking at Fig. R5/ Fig. 5 we can compare how centroparietal activity unfolds in the visual and audiovisual conditions.

In both conditions, the activity in centroparietal sensors peaked before < 500 ms after the stimulus onset. Then, the centroparietal activity remains stable until the response phase. This suggests that participants generally commit to their final perceptual decision in the first 500 ms. Hence, the sustained activity profile suggests that the decision information is maintained until the response phase. We must note that this univariate analysis only provides information about the temporal profile of evidence accumulation and does not allow to infer which information is stored in centroparietal activity nor to discriminate between different dynamical information processing models. Therefore, the decoding analyses in combination with TG are more informative in that regard, as they allow to quantify how dynamical/stable in time are the activity patterns associated to certain stimulus representations.

9. A more detailed description of the statistical analysis, particularly the key cluster-based permutation analysis, would be helpful for the reader and necessary to be able to reproduce it. The presented paragraph at the very end of the methods is rather short given such a complex and important analysis. It leaves the reader wondering how reported p values were obtained.

Usually cluster t-values of the permutation distribution are used to define data-driven significance thresholds as in the original publication by Maris and Oostenveld (2007). These clusters would be obtained in space and time to correct for multiple comparisons. Here, it seems that the authors only corrected for multiple comparisons in the spatial domain but not the temporal domain (i.e., number of consecutive significant temporal samples). This may not be a flaw of the analysis but more of the description. However, if the temporal domain wasn't considered this should also be included in the cluster analysis. I'd invite the authors to clarify this part of their analysis.

We apologize for being unclear here. We have added more details about the procedure in the methods section:

“Positive and negative clusters were then formed separately by grouping temporally adjacent data points exceeding a threshold corresponding to $\alpha = 0.05$ (two-tailed). Then, cluster-level statistics were calculated using the 'weighted cluster mass' method, a statistic that combines cluster size and intensity (Hayasaka & Nichols, 2004). To obtain a null distribution of this cluster-level statistic, we repeated the same process 1000 times permuting trial-condition associations in each iteration. This yields cluster p-values, corrected for multiple comparisons. The null hypothesis was rejected for those clusters for which the p-value were smaller than 0.05, compared to the null distribution of cluster-level statistics.”

We would like to clarify that in our TFM, cluster-based permutation analyses were performed in a 2D space (but instead of “time x space”, or “time x frequency”, we considered “time_{training} x time_{decoding}”). This is now clarified in the main text: “This procedure controls for multiple comparisons across training and decoding time combinations.” The spatial dimension was not directly involved in statistical comparison. In order to compute a TFM, we already aggregate across the spatial dimension by performing multivariate classification analysis. Independent of the number of sensors that go into this analysis, the statistical comparison always involves the single TFM (time X time) that it produces, so no correction across space is needed.

10. Figure captions of figures 3-8 state that significant clusters are depicted by thin red markings. While these are very hard to see they also appear patchy and small in their extent. Following my previous point, this might suggest that they would not survive correction for multiple comparisons in the spatial and temporal domain. The temporal windows described in the text are not presented in the decoding figures, their addition (as in S11) might help the reader. Please disregard this comment if this is merely an issue with visibility of the markings and consider making them more visible.

Thanks for pointing this out. The reported clusters are not small in the extend. Actually, there are not usually more than 2 or 3 (big) clusters in each TGM, but they might produce the patchy impression because of some non-significant isolated pixels within the clusters. Following the reviewer’s suggestion, we have increased the thickness of the contour lines to improve the clusters visibility. As noted in the previous comment, multiple comparisons are fully controlled for in the two time dimensions (training X testing time), and not applicable in the spatial dimension.

11. Some result statements about the absence of an effect in these data (i.e., tests that yielded a p value above the arbitrary frequentist significance threshold of $p = .05$) cannot be made with such strengths using the frequentist framework (e.g., line 143-144 and 147-149). The reason being that absence of evidence is not evidence of absence. To support these statements, I’d suggest that the authors use Bayesian statistics and Bayes Factors to be able to make statements on the likelihood of their data under the null hypothesis (absence of an effect).

We thank the reviewer for this suggestion. We have now added Bayes Factor statistics in those contrasts in which we found null effects.

12. The discussion would benefit from discussing the link between the temporal results and of the decoder and their functional interpretation in more detail. It would also benefit from discussing what the effects of other auditory stimuli might have been, as bottom-up sound-driven modulations of early visual cortex are a key finding. That is, would more complex sounds trigger activity in other parts of the visual cortex further up the hierarchy (e.g. extrastriate cortex) and not in early visual cortex?

We believe that the temporal interpretation of the decoder results is already addressed in:

- 1) *Dynamical models comparison*: *The proposed models are inspired on different hypotheses about information processing and formulate different predictions considering the latency and temporal persistence of the audiovisual interactions (Fig. 10).*

“Our results are better explained by a “late maintenance” model in which stimulus information after being encoded in early sensory regions is remapped as a decision variable (DV) and stored until the response phase. According to this model’s predictions, sounds would improve visual sensitivity by enhancing the maintenance over time of task-relevant visual information encoded at 500 ms after the stimulus onset (see that the target information encoded at 500 ms was highly correlated with participants sensitivity).”

- 2) *The response to point 7*: *Here we offered a functional interpretation of the TG decoding results considering the latency of the significant clusters and the ROI in which they are localized (see also Fig. S12).*

In relation to the question “what the effect of more complex auditory stimuli in the brain would be”, we have added the following paragraph: “Previous research also showed that complex natural sounds also evoke category-specific information that can be read-out from early visual regions (Vetter et al., 2014). Similarly, auditory motion can activate higher order visual regions like occipito-temporal cortex (hMT+, Poirier et al., 2005; Rezk et al., 2020). All these results together highlight that audiovisual interactions are pervasive in the human brain. Nevertheless, the neural mechanisms by which simple and complex auditory stimuli are feedbacked to striate and extrastriate visual areas might be qualitatively different and further research is required to address this question.”

Minor concerns

What was the motivation for choosing a set of very bright/salient colours for experiment 2? For example, magenta might be more salient than yellow. I presume it’s unlikely that the order of the colours might have biased the results substantially, but I would ask the authors to clarify how the colour sequence was picked and randomized for each trial.

In the second experiment, although the fixation point changed its color three times in each trial, they were matched in luminance with the “normal” fixation point to avoid large asymmetries in saliency. Moreover, it is unlikely that the colors could bias participant in any systematic way as the color sequence was randomly generated for each trial (e.g. Cyan-Blue-Yellow; Blue-Yellow-Cyan; Yellow-Blue-Blue). We now describe this in more detail in Methods:

“The order of the three fixation point color changes was randomized in each trial and they barely differed in luminance (405 +- 30 cd/m²).”

The authors should consider showing the behavioural results of experiment 2 in the main text and not only as a supplementary figure (currently S3), since the absence of behavioural effects given the contrast level and modality is an important result for their claims. Further, results depicted in figure S5 and S6 are being mentioned in line 179 as prior evidence. I think these are deserving of being incorporated in a figure of the main manuscript.

We have included the behavioral results of experiment 2 in the main text. We have also included the correlation analyses depicted in S5 and S6.

The supplement shows two figures labelled S5.

Corrected.

Please use the same scale for all dprime and all criterion temporal generalization plots in figs 4-7. Currently the colourmaps are varying in range.

We have scaled the colormaps ranges across similar TGM analyses.

Line 147: please specify more clearly what a “more liberal response threshold” actually means in the context of your task.

We have rephrased the sentence: “This way, lower c parameter values represented a larger bias to report the signal as present”

Line 347: It would be informative to report accuracy values alongside dprime.

Done.

Line 612: How many participants or blocks required an adjustment due to ceiling or floor effects? Please add this information, if available.

Participants have an overall tendency to decrease the performance as the experiment progressed Fig R6. That required to adjust the contrast threshold in 31% (SD = 1.21) and 32 % (SD = 1.11) of the blocks in the center and periphery visual field conditions respectively. Contrast threshold adjustments resulted on an average of 70% of accuracy in both visual field conditions (Fig. 1A). This information has been added to the methods section.

Figure R6 Average over participants contrast threshold as a function of the block number for the center and periphery visual fields.

I outline several suggestions for phrasing and typos:

We sincerely appreciate the detailed and thorough manuscript revision. We have corrected the reported typos.

- Line 64: Most previous neuroimaging (delete of)

Done

- Line 115: This is in contrast

Done

- The sentence in line 207 is truncated, please double check the ending

This seemed to be a problem with pdf conversion. Now it is fixed.

- Line 251: information conformed to a continuous pattern

Done

- Line 268: the word between is redundant

Done

- Line 398: false alarm reports

Done

- Line 457: their feature-based attention

Done

- Line 502: the percept of an additional

Done

- I believe the duration in line 629 should read 33ms instead of 32ms.

Done

- 706: involved in short-term memory

Done

Reviewers' comments:

Reviewer #2 (Remarks to the Author):

The revised manuscript appears to be in a much-improved state. However, few minor points remain that I would like the authors to address before I can recommend its publication.

Point 5 (lines 290 ff.): I appreciate your efforts in characterising the evidence accumulation process in the form of the CPP component more thoroughly. The added section is a reasonable way to address and corroborate the manuscript's claims in this regard. Nevertheless, I have the following additional suggestions:

- Please scale the x-axis of Fig. 5B and C similarly in space, particularly, since Fig. 5C directly refers to the content and time points presented in 5B. That is, covering the same horizontal space.
- Were the significant temporal clusters obtained using a cluster correction for multiple (temporal) comparisons?
- What does the sharp centro-parietal activity decrease around 250 ms post-stimulus onset indicate exactly? If crossmodal information is simply not available or not enhanced by attention yet, why would there be a decrease compared to the visual condition? If, as suggested by the authors, audiovisual interactions only start to act on evidence accumulation post 350 ms, is there some form of a sensory suppression or inhibition effect going on prior to the onset of this evidence accumulation? You have hinted at the interpretation of evidence accumulation only starting around 350 ms and being enhanced in the audiovisual condition in your addition to the results section but in my opinion this very relevant interpretation, which is fitting the study's research questions well, could be phrased more explicitly in this amended results section.
- In the rebuttal to my fifth point, I have difficulties to understand the last sentence presented under "Discussion". I'd suggest splitting this sentence into two to enhance clarity.

Line 156 and 755: shouldn't a criterion closer to 0 (the zero-bias point, as in the cited book, ref 95) indicate less bias (i.e., in the audiovisual condition)? The SDT analysis appears to be set up in a "traditional" way for yes-no tasks, which would speak to this interpretation. If I'm wrong, please correct me and clarify this aspect in the methods section.

Line 586: While feedbacked is an existing word, it might be worth considering to rephrase this as "is fed back to".

Reviewer #4 (Remarks to the Author): (Replacement for the previous Reviewer #3)

This is an interesting study on the cerebral mechanisms underlying the sound-driven enhancement of visual perception. The study has several technical strengths and does offers novel data and insights, though the results are also less specific than the authors seem to claim.

Overall I find that the statistical reporting lacks detail and the interpretation of the results in the light of previous work can be improved.

General comments:

In their revision the authors have done a decent job in addressing many of the reviewer comments, though some remain. As far as I can tell I did not review the previous version, and I read the manuscript before the reply letter. Yet, I found a number of concerns that align with the previous reviewer comments.

The authors ask the question of whether sounds enhance visual sensitivity at early (i.e. sensory encoding), late (i.e. decision formation) or both perceptual stages. In line with other studies the current data here point to a distributed origin that among other mechanisms may involve a higher accumulation rate of relevant evidence during decision making and an enhanced maintenance of information in working memory.

The authors also phrase two further questions in the introduction, which I found less clear: i) whether any sound-induced increase in false alarms reflects a decisional-level bias or it also depends on changes in 'perception', and ii) whether 'multisensory integration' is automatic. While I

believe to understand what the authors mean I find the terms perception and multisensory integration here problematic. First, does perception not refer to the process of sensory information reaching awareness? Awareness here is only measured indirectly via an explicit task, and any decision-level bias is a phenomenon only occurring in an explicit task, hence not comparable to awareness. And multisensory integration, is a term that many use to refer to weighted combination of redundant signals, i.e. the explicit fusion of two estimates of equivalent sensory features (e.g. two spatial location estimates) that can be e.g. averaged on a common axis. The current paradigms probably tap into other multisensory processes / interactions / phenomena, but using the term integration here may be confusing.

This brings us to a key point: When interpreting the results, I was wondering what to make of the rather small effect in parietal cortex (Fig. 4). A critical feature may be the difference in task, which is not proper multisensory integration but rather a visual detection task. Many studies on integration or multisensory causal inference have reported strong signatures of integration in ppc (e.g. Rohe & Noppeney several papers, Cao et al Neuron 2019). In contrast, the visual + additional multisensory influence paradigm probed here seems to largely engage visual + temporal visual regions in the multisensory aspect. This difference between integration and basic signal enhancement by an auxiliary stimulus is worth commenting on.

Another key point is the reporting of statistical results. I understand that most results are obtained from cluster-based permutation procedures. Here, I find the used 1000 permutations on the low end and the first-level threshold is very lenient with $p < 0.05$ (see e.g. Eklund et al PNAS 2016). What are the reasons for these specific choices? Further, the authors report only p-values but no effect sizes, which make it impossible to compare the effects between ROIs or analyses and makes it difficult to assess them against the null results in Experiment 2. For example, the authors could report for each cluster (or attempted test that does not yield a cluster) the max / min t-value within the cluster or tested region. For the correlation analysis, was this a spearman's correlation? The text states that 'we tested the correlation distribution differed from 0.' You mean the mean differed from zero? Comparing a distribution against a single number makes no sense to me. An improved reporting of effect sizes seems important in the light of the authors claim that Experiment 2 returned no effects.

One technical concern I have is about the source activity shown in Figure R3. This shows a much stronger ERF in the red (AV) condition at very short latencies across all ROIs. Given that the main difference to the green trace must be the auditory stimulus, I assume this reflects and auditory evoked response that spreads to all ROIs, in particular VC. In line with this, the authors interpret this as sign of early multisensory influences. But how can we be sure that this is indeed a neural response and not simply current spread from auditory regions? Not showing data from an auditory ROI (as reviewer 2 requested) does not help. Also, the authors used 10% smoothing for the source analysis, which enhances potential blurring of signals in my understanding. This current spread in principle may not be an issue for the decoder, but it may also affect the effective SNR of visual information in the two conditions. This difference in SNR may affect e.g. classification bias and may also relate to the asymmetry of the TGM analysis in Figure 4 when training the decoder on both the AV and V conditions.

Finally, I was wondering what to make of the temporal specificity of the neuro-behavioral correlation in Figure 3 and that of the multisensory effects in Figure 4. Does it matter that the correlation with behavior for d' is near zero in the time-window of the significant cluster for VC in Figure 4b? The authors seem to take Figure 3 as showing that the decoding approach in general mimics behavior, but this is certainly not the case. Some commenting on the different temporal profiles would be important in my view.

Details:

L 77f. For the discussion of multisensory processes in the face of attention or awareness I find the work by Uta Noppeney's group highly relevant (e.g. Delong and Noppeney for awareness and Noppeney and Zuanazzi for attention).

L 97. The precise choice of the four ROIs seems to be derived from the previous literature. Still, I was wondering why the authors focused on DLFPFC and not inferior temporal gyrus, a region implied by many studies as hub for multisensory conflict resolution.

L. 277. The authors here conclude on the interplay of different ROIs, but did not quantify so.

Reviewer #5 (Remarks to the Author): (Replacement for the previous Reviewer #1)

The study by Pérez-Bellido et al. explores the effect of auditory stimuli to visual perception. They use a decoding approach to extract SDT sensitivity and criterion parameters from MEG data and compare them to the behavioral data. In my opinion, the authors have satisfactorily responded to the comments by two previous reviewers.

I have some minor questions left:

- Figure R1A/S4A: Is plot R1A/S4A for one subject like the B column? If yes, could you say in the figure legend?
- Figure R1B/S4B: It should be mentioned what are the 6 points per color in the plots. I believe they are for 6 experimental blocks? Also can you discuss the fact that the behavioural values are much lower than the decoder values?
- Figure R2: what is the unit for activity modulation?
- Figure 3 in the main text: Panel 3B seems to be not mentioned or referred in the main text at all. It is a bit difficult to understand. What do solid and dashed lines present?
- Figure 5 in the main text: Is this illustrating an average across subjects or a single subject?
- In Methods: "Therefore, we again applied SDT to calculate participants sensitivity, but assuming a same-different "independent observation" model paradigm." It is unclear what they mean by "independent observation" model paradigm.

Reviewers' comments:

Reviewer #2 (Remarks to the Author):

The revised manuscript appears to be in a much-improved state. However, few minor points remain that I would like the authors to address before I can recommend its publication.

We appreciate the reviewer's work in helping to improve the manuscript. We hope that now the manuscript is ready for publication.

- 1) Point 5 (lines 290 ff.): I appreciate your efforts in characterising the evidence accumulation process in the form of the CPP component more thoroughly. The added section is a reasonable way to address and corroborate the manuscript's claims in this regard. Nevertheless, I have the following additional suggestions: Please scale the x-axis of Fig. 5B and C similarly in space, particularly, since Fig. 5C directly refers to the content and time points presented in 5B. That is, covering the same horizontal space.

Thanks. We have updated the figure following the reviewer's suggestion. We have realized that there was a mistake in Fig. 5B (the size of the significant cluster in the visual condition) that is now corrected.

- 2) Were the significant temporal clusters obtained using a cluster correction for multiple (temporal) comparisons?

Indeed, we have applied cluster-based permutations statistics in all the analyses involving multidimensional data. We have added this information to the results section:

"Using cluster-based permutation tests (correcting for multiple comparisons across time) we found that whereas in the visual condition, the processing of the visual target induced a broadly spatially distributed activity pattern across the ..."

- 3) What does the sharp centro-parietal activity decrease around 250 ms post-stimulus onset indicate exactly? If crossmodal information is simply not available or not enhanced by attention yet, why would there be a decrease compared to the visual condition? If, as suggested by the authors, audiovisual interactions only start to act on evidence accumulation post 350 ms, is there some form of a sensory suppression or inhibition effect going on prior to the onset of this evidence accumulation? You have hinted at the interpretation of evidence accumulation only starting around 350 ms and being enhanced in the audiovisual condition in your addition to the results section but in my opinion this very relevant interpretation, which is fitting the study's research questions well, could be phrased more explicitly in this amended results section.

We thank the reviewer for this interesting suggestion. The centroparietal activity suppression taking place at around 350 ms is also informative and deserves to be discussed in the manuscript.

“Interestingly, we found that in the audiovisual conditions, the centroparietal activity is drastically reduced at around 350 ms, suggesting that the ongoing process of evidence accumulation is reset. Such reset might correspond to a strategic reorienting of processing resources towards those sensory samples that more likely encode the visual target, that indeed manifests subsequently by a strong increase in evidence accumulation.”

- 4) In the rebuttal to my fifth point, I have difficulties to understand the last sentence presented under “Discussion”. I’d suggest splitting this sentence into two to enhance clarity.

We have improved the clarity of this paragraph by splitting it in two.

“Future studies should further investigate whether the heterogenous auditory-to-visual cortex connectivity pattern reported in monkeys also exists in humans (Beer et al., 2011). If it does, the next question to answer is whether it plays a functional role for multisensory processing, as proposed in previous studies (Jaekl & Harris, 2009; Jaekl & Soto-Faraco, 2010).”

- 5) Line 156 and 755: shouldn’t a criterion closer to 0 (the zero-bias point, as in the cited book, ref 95) indicate less bias (i.e., in the audiovisual condition)? The SDT analysis appears to be set up in a “traditional” way for yes-no tasks, which would speak to this interpretation. If I’m wrong, please correct me and clarify this aspect in the methods section.

As correctly pointed out by the reviewer, a $c = 0$ indicates that participants are not biased either to report more signal-present than signal-absent trials (or vice versa). Here we find that in both visual and audiovisual conditions, $c > 0$, meaning that participants are generally biased to report signal-absent trials. However, $c_{AV} < c_V$, indicates that the bias to classify the trials as signal-absent is lower in the audiovisual conditions. That is, in the audiovisual condition they experience fewer misses but more false alarms than in the visual condition. This is now clarified in the main text:

“Furthermore, the criterion (c) parameter was larger than 0 in both the visual and audiovisual conditions, manifesting an overall bias to classify the trials as signal-absent. However, as predicted, this bias was lower in the audiovisual compared to the visual conditions ($c = 0.56$ vs 0.79 ; $F_{1,23} = 23.97$, $P < 0.001$, $\eta = 0.014$; Fig. 2D), indicating that participants were more likely to report signal-present in the audiovisual trials.”

- 6) Line 586: While feedbacked is an existing word, it might be worth considering to rephrase this as “is fed back to”.

Thanks for the suggestion. We have updated the text accordingly.

Reviewer #4 (Remarks to the Author): (Replacement for the previous Reviewer #3)

This is an interesting study on the cerebral mechanisms underlying the sound-driven enhancement of visual perception. The study has several technical strengths and does offer novel data and insights, though the results are also less specific than the authors seem to claim.

Overall I find that the statistical reporting lacks detail and the interpretation of the results in the light of previous work can be improved.

We thank the reviewer's overall positive assessment of the manuscript. We have carefully assessed all her/his recommendations to improve our work.

General comments:

In their revision the authors have done a decent job in addressing many of the reviewer comments, though some remain. As far as I can tell I did not review the previous version, and I read the manuscript before the reply letter. Yet, I found a number of concerns that align with the previous reviewer comments.

The authors ask the question of whether sounds enhance visual sensitivity at early (i.e. sensory encoding), late (i.e. decision formation) or both perceptual stages. In line with other studies the current data here point to a distributed origin that among other mechanisms may involve a higher accumulation rate of relevant evidence during decision making and an enhanced maintenance of information in working memory.

- 1) The authors also phrase two further questions in the introduction, which I found less clear: i) whether any sound-induced increase in false alarms reflects a decisional-level bias or it also depends on changes in 'perception', and ii) whether 'multisensory integration' is automatic. While I believe to understand what the authors mean I find the terms perception and multisensory integration here problematic. First, does perception not refer to the process of sensory information reaching awareness? Awareness here is only measured indirectly via an explicit task, and any decision-level bias is a phenomenon only occurring in an explicit task, hence not comparable to awareness.

The use of the term 'perception' here was meant to label one possible level at which a sound-induced visual bias might manifest, and intended solely to be contrasted with 'decision'. It might be the case that the presence of a sound causes people to actually see a concurrent light flash more often, even when it is absent. We would consider that a perceptual-level bias. In contrast, it might also be the case that the presence of a sound causes people to push a button more often, without any change in actual perception. We agree with the reviewer that, given the use of an explicit task, we cannot distinguish between these two alternatives based solely on the behavioural data. Therefore, instead, we aim to do so based in part on the neural data.

To avoid any confusion, we now avoid the use of a broadly unspecific term like a “change in perception”, and instead will describe it as a perceptual-level bias (contrasted with decisional-level bias):

“Thus, whether the sound-induced increase in false alarms in visual detection tasks exclusively represents a decisional-level bias or it might also reflect a perceptual-level bias is still unknown.”

- 2) And multisensory integration, is a term that many use to refer to weighted combination of redundant signals, i.e. the explicit fusion of two estimates of equivalent sensory features (e.g. two spatial location estimates) that can be e.g. averaged on a common axis. The current paradigms probably tap into other multisensory processes / interactions / phenomena, but using the term integration here may be confusing.

We agree with the reviewer’s point of view. We did not mean to use the term ‘multisensory integration’ to necessarily entail integration of information on e.g. a common axis.

We have revised the manuscript substituting the term ‘integration’ with more precise terminology in each instance. For example:

Instead of:

“Finally, another disputed question taps into the automaticity of multisensory integration.”

Now:

“Finally, another disputed question taps into the automaticity of multisensory interplay.”

- 3) This brings us to a key point: When interpreting the results, I was wondering what to make of the rather small effect in parietal cortex (Fig. 4). A critical feature may be the difference in task, which is not proper multisensory integration but rather a visual detection task. Many studies on integration or multisensory causal inference have reported strong signatures of integration in ppc (e.g. Rohe & Noppeney several papers, Cao et al Neuron 2019). In contrast, the visual + additional multisensory influence paradigm probed here seems to largely engage visual + temporal visual regions in the multisensory aspect. This difference between integration and basic signal enhancement by an auxiliary stimulus is worth commenting on.

We agree with the reviewer: We do not believe that the sound-induced enhancement in information decoding that we identified in the parietal cortex ROI is related in any manner to the parietal activity reported by Rohe & Noppeney, 2016 / Cao et al, 2019. That would indeed explain the small effect that we found in the in the parietal ROI.

Previous research on perceptual decision making has found that the activity in parietal regions is strongly correlated to the process of evidence accumulation. Thus, considering that our task involves the accumulation of perceptual evidence, the significant cluster is likely indexing a punctual enhancement in evidence accumulation in the audiovisual trials. As suggested by the reviewer, we now comment on this in the main text:

“This interpretation is supported by the sound-induced enhancement in information decoding in the parietal cortex (Fig. 4B). In contrast with previous multisensory research that characterized the activity in parietal regions as a signature of multisensory integration (Cao et al., 2019; Rohe & Noppeney, 2016), in the light of our visual detection paradigm it might better reflect a change in the process of evidence accumulation (O’Connell et al., 2012; Ratcliff et al., 2009).”

- 4) Another key point is the reporting of statistical results. I understand that most results are obtained from cluster-based permutation procedures. Here, I find the used 1000 permutations on the low end and the first-level threshold is very lenient with $p < 0.05$ (see e.g. Eklund et al PNAS 2016). What are the reasons for these specific choices?

We would like to clarify our approach to avoid misunderstanding. First, regarding the choice of the cluster-forming threshold. Selecting $\alpha = 0.05$ is a standard cluster-forming threshold in the field. Importantly, the choice of this threshold has no effect on the false alarm rate of the test (cite Maris & Oostenveld), and is independent of the choice of the significance level (unhelpfully both are typically referred to by ‘alpha’). We are confident in the use of permutation tests to assess the significance of the reported effects. In fact, the paper mentioned by the reviewer (Eklund et al. 2016) argues in favour of the use of permutation tests, given their robustness to the lack of data compliance to statistical assumptions required in other parametrical statistics (Fig. R1).

R1. Figure 1 copied from Eklund et al. 2016 comparing different cluster-based statistical tests: Using a significance level of $\alpha = 0.05$, only the permutation tests kept FWE/Type 1 error within the expected boundary (the analyses shown here were performed before the different benchmarked toolboxes (AFNI, SPM and FSL) corrected their statistical inference methods).

The decision of using 1000 permutations also reflects standard practice. We emphasize that the choice here only influences the resolution with which one can estimate p-values, and that the choice of 1000 permutations should yield a resolution of 0.001. The recommended approach by FieldTrip’s documentation is to use 500 permutations or double it for a more

fine grained P value calculation: *“In this tutorial, we use `cfg.numrandomization = 100`. This number is too small for actual applications. As a rule of thumb, use `cfg.numrandomization = 500`, and double this number if it turns out that the p-value differs from the critical alpha-level (0.05 or 0.01) by less than 0.02.”* See

https://www.fieldtriptoolbox.org/tutorial/cluster_permutation_timelock/

Our resultant p-values were sufficiently different from the critical alpha (significance) level that there was no need to increase the number of permutations, by this guideline. Thus, we believe that the use of 1000 permutations remains within the recommended range of permutations. To be absolutely certain, we have now run the same analyses using 10000 permutations, and can conclude that the results were virtually identical to the ones that we have already reported in the manuscript. Therefore, we have decided to maintain the figures that were initially generated using 1000 permutation.

- 5) Further, the authors report only p-values but no effect sizes, which make it impossible to compare the effects between ROIs or analyses and makes it difficult to assess them against the null results in Experiment 2. For example, the authors could report for each cluster (or attempted test that does not yield a cluster) the max / min t-value within the cluster or tested region.

In the previous version of the manuscript we already provided η^2 and Bayes Factor for significant and non-significant parametric statistical analyses. Note that cluster-based permutation analyses belong to the family of non-parametric statistics and therefore effect sizes cannot be computed straightforwardly. As the reviewer suggests, we now report the maximum t-value within the most significant clusters in each analysis and its cluster-level t-value.

In methods section we have also included the following sentence:

“In the results section, for each independent analysis we report the $t_{max/min}$ statistics within the most positive/negative significant clusters and its corresponding cluster-level t-value.”

- 6) For the correlation analysis, was this a spearman’s correlation? The text states that ‘we tested the correlation distribution differed from 0.’ You mean the mean differed from zero? Comparing a distribution against a single number makes no sense to me.

We used a Pearson correlation. This information is now available in the manuscript.

We have also corrected the inaccurate statement:

“Using a cluster-based permutation approach we tested in which decoding train-test combinations of the TGM the mean of the group-level distribution of correlation coefficients was significantly different from 0.”

- 7) One technical concern I have is about the source activity shown in Figure R3. This shows a much stronger ERF in the red (AV) condition at very short latencies across all ROIs. Given that the main difference to the green trace must be the auditory stimulus, I assume this reflects an auditory evoked response that spreads to all ROIs, in particular VC. In line with this, the authors interpret this as sign of early multisensory influences. But how can we be sure that this is indeed a neural response and not simply current spread from auditory regions? Not showing data from an auditory ROI (as reviewer 2 requested) does not help. Also, the authors used 10% smoothing for the source analysis, which enhances potential blurring of signals in my understanding. This current spread in principle may not be an issue for the decoder, but it may also affect the effective SNR of visual information in the two conditions. This difference in SNR may affect e.g. classification bias and may also relate to the asymmetry of the TGM analysis in Figure 4 when training the decoder on both the AV and V conditions.

We would like to clarify that we do not interpret the absolute difference in ERF activity between the visual and the audiovisual conditions in visual cortex as evidence for early multisensory interaction. In the second experiment we found that sounds induced an activity pattern in visual cortex similar to the activity evoked by an actual visual stimulus. This interaction is compatible with early crossmodal interaction, nevertheless, as we previously mentioned, this interpretation is exclusively based on multivariate decoding analyses. Although it is likely that sound-induced ERFs modulation registered in most of the ROIs (Fig. S11 is now added to supplementary materials) might actually correspond to a crossmodal neural modulation, the reviewer is right in pointing out that it is difficult to tease apart an actual neural modulation from auditory activity spreading. Thus, we avoid to interpret the mentioned absolute differences in ERF as a signature of early crossmodal interaction. Now we discuss this issue in the manuscript:

“It is important to note that whereas we replicated that sounds induced univariate activity modulations in most of our selected ROIs and specially in the visual cortex (Fig. S11), we must be cautious in interpreting them as genuine crossmodal sensory modulations. Despite the spatial filtering applied to the ERF signal using beamforming source reconstruction, auditory activity can potentially spread from the auditory cortex to the virtual channels in adjacent brain regions, including the visual cortex. Thus, the use of multivariate decoding is more accurate in discriminating visual-stimulus specific activity patterns from auditory trials regardless overall changes on univariate activity”.

In relation to the decoding asymmetry, it is important to recall that the main results are replicated even when the decoder is trained independently in the visual and audiovisual trials. We also explored whether differences in univariate activity might be at the base of the decoding bias in the second experiment (Fig. 9B), but it does not seem to be the case (Fig. S12). For completeness, in our previous revision and the supplementary materials (Fig. S1B) we shared the results of source localization analyses for the auditory activity. We also performed an auditory stimulus decoding analysis in the auditory ROI (Fig. R4).

Figure R2. A. Visual (0.2 to 0.3) and B. auditory (0.1 to 0.2 s) source-reconstructed activity projected in a surface model of the brain. Same results are plotted in a standard MNI template for the visual C and the auditory D activities. See that source-reconstructed signal in V1 (striate cortex) is also modulated by auditory stimulation.

- 8) Finally, I was wondering what to make of the temporal specificity of the neuro-behavioral correlation in Figure 3 and that of the multisensory effects in Figure 4. Does it matter that the correlation with behavior for d' is near zero in the time-window of the significant cluster for VC in Figure 4b? The authors seem to take Figure 3 as showing that the decoding approach in general mimics behavior, but this is certainly not the case. Some commenting on the different temporal profiles would be important in my view.

The goal of Fig. 3C analysis is to validate whether the decoded parameters are informative about participants behavior (and at which time points). Therefore, in Fig. 3C (upper panels) we correlated the behavioral d' (taking the visual and in the audiovisual conditions together) with the decoded d' (taking the visual and in the audiovisual conditions together). Instead, in Figure 4B we represent the *difference* between the decoded d' in the audiovisual and visual conditions. Thus, the two analyses cannot be directly compared. Please note, however, that according to the correlation analysis, the information neurally represented at around 500 ms is the one that encodes most of the information that is relevant for the perceptual decision (Fig. 3C and Fig. S3). Importantly, it is the reactivation of these same specific representations (but at later latencies) what gets enhanced in the audiovisual compared to the visual conditions. Now we comment on this in the manuscript:

“According to this model’s predictions, sounds would improve visual sensitivity by enhancing the maintenance over time of visual information encoded at 500 ms after the stimulus onset. The sound-induced enhancement of the information encoded at these specific time-points does not seem like a coincidence, as 500 ms neural activity appears important for the perceptual decision itself (Fig. 3C)”

9) Details:

L 77f. For the discussion of multisensory processes in the face of attention or awareness I find the work by Uta Noppeney’s group highly relevant (e.g. DeLong and Noppeney for awareness and Noppeney and Zuanazzi for attention).

We thank the reviewer for this suggestion, and have updated the manuscript with citations to these two relevant studies.

L 97. The precise choice of the four ROIs seems to be derived from the previous literature. Still, I was wondering why the authors focused on DLPFC and not inferior temporal gyrus, a region implied by many studies as hub for multisensory conflict resolution.

In this study we are specifically interested in investigating whether sounds interact with visual detection. Perceptual decision making and visual detection rely on a brain network encompassing visual cortex, parietal cortex, inferotemporal cortex and dorsolateral prefrontal cortex. The selection of these brain regions is inspired by previous research on the field of visual detection (King et al., 2016; Sergent et al., 2011, 2013; van Vugt et al., 2018). In this regard, we believe the inclusion of the DLPFC is justified given its critical role on visual detection awareness and more generally in decision making processes (Kim & Shadlen, 1999; Philiastides et al., 2011; van Vugt et al., 2018).

Our experimental task was not designed to elicit multisensory *conflict*, so we believe inclusion of regions involved in multisensory conflict resolution like the IFG is not a priori the obvious choice to make. Given the above, we decided to focus on DLPFC and the other regions included.

L. 277. The authors here conclude on the interplay of different ROIs, but did not quantify so.

This is a very interesting but complex question to address. We are afraid that even (unisensory) research on visual detection has not been able to figure out how the different brain regions interact to process visual information. Thus, we are not that ambitious in our goals and by the moment we leave this question out of the scope of the current manuscript. We have formulated the reviewer's proposal as another open question to the discussion:

"In the future, the combination of TG decoding with more specific functional and anatomical localizers will help to understand with a greater level of detail how the different brain regions collaborate to enhance visual detection when an auditory stimulus is presented, and how specifically visual information is transformed across the multiple stages that conform the perceptual decision-making hierarchy."

Reviewer #5 (Remarks to the Author): (Replacement for the previous Reviewer #1)

The study by Pérez-Bellido et al. explores the effect of auditory stimuli to visual perception. They use a decoding approach to extract SDT sensitivity and criterion parameters from MEG data and compare them to the behavioral data. In my opinion, the authors have satisfactorily responded to the comments by two previous reviewers.

Thank you for the positive assessment of our manuscript revision.

I have some minor questions left:

- 1) Figure R1A/S4A: Is plot R1A/S4A for one subject like the B column? If yes, could you say in the figure legend?

We have added the missing information:

“Figure S4.A Group level temporal generalization matrixes with d' and c correlation values in the first experiment. Blue and red crosses indicate the two time points of interest (0 to 0.5 s) used in the single-subject correlation analysis showed in B. B Example of d' and c correlation analysis (upper and bottom panels respectively) for one representative subject at two different time points.”

- 2) Figure R1B/S4B: It should be mentioned what are the 6 points per color in the plots. I believe they are for 6 experimental blocks?

Thank you. We have specified this more clearly in the text. “Each point represents the SDT parameter estimates calculated for each experimental block ($N = 6$).”

- 3) Also can you discuss the fact that the behavioural values are much lower than the decoder values?

The result in the figure indicates the opposite. For instance, in S4.B for a behavioral d' of nearly 2.8 units, the decoded d' is nearly 0.1. This is an extreme example, however in most of the cases the behavioral d' s are higher than the decoded ones. See that group level behavioral d' is around 1.5 and group-level decoded d' is around 0.6. Finding lower decoded d' than behavioral d' it is expected given that behavior is a direct measure of participants ability to discriminate the targets whereas the decoded d' relied on an indirect measure of participants sensitivity that depends on activity measured at multiple MEG channels possibly corrupted with other noise sources.

- 4) Figure R2: what is the unit for activity modulation?

The units are “% of signal change”. We have added this information to the supplementary figure (Fig. S1).

- 5) Figure 3 in the main text: Panel 3B seems to be not mentioned or referred in the main text at all. It is a bit difficult to understand. What do solid and dashed lines present?

Now Figure 3B is referred in the multivariate analysis methods section. We have also improved the description of the different elements conforming Fig.3B panel.

“B. A multivariate pattern classifier was trained on visual and audiovisual (green and red simulated points respectively) conditions together to optimally discriminate between S+ and S- trials (empty and full circles respectively). The trained classifier was cross validated on

visual and audiovisual trials separately. The solid and dashed lines represent the probability density functions of the S+ and S- trials distributions as a function of the discriminant channel output. Specifically, trials with a discriminant channel output larger/smaller than 0 (vertical line) are classified as signal-present/absent. We categorized S+ trials classified as signal-present as hits, and the S- trials classified as signal-present as false alarms. Note that although a classifier might be equally good in discriminating S+ from S- trials in all the conditions, there might be systematic biases to classify more often the trials in the audiovisual condition (middle panel) as signal-present than in the visual condition (bottom panel)."

- 6) Figure 5 in the main text: Is this illustrating an average across subjects or a single subject?

We have now detailed this in the main text:

"Figure 5 A Group-level difference between the sensor-level ERF activity in the signal-present and signal-absent conditions measured at (A) whole-brain scalp topography, and (B) centroparietal sensors (red points in the topography inset)."

- 7) In Methods: "Therefore, we again applied SDT to calculate participants sensitivity, but assuming a same-different "independent observation" model paradigm." It is unclear what they mean by "independent observation" model paradigm.

With 'independent observation' model paradigm we referred to what is essentially the standard in signal detection theory (Macmillan & Creelman, 2004). We appreciate that this undefined term may have caused confusion and have now removed it.

We have updated the text:

"Therefore, we again applied SDT to calculate participants sensitivity. However, given that participants had to compare whether the fixation point colors were same or different, we modelled the SDT parameters assuming a "same-different" decision paradigm (Macmillan & Creelman, 2004)."

REVIEWERS' COMMENTS:

Reviewer #2 (Remarks to the Author):

The authors have satisfactorily addressed all my concerns. From a scientific standpoint I can recommend publication of this manuscript. However, I'd urge them to proofread the manuscript thoroughly, as multiple instance of doubled words, a truncated sentence, and some random letters remain present.

Reviewer #4 (Remarks to the Author):

The authors have provided a solid revision and I'm happy with their replies to my comments.